# Struct-XLM: A Structure Discovery Multilingual Language Model for Enhancing Cross-lingual Transfer through Reinforcement Learning

**Linjuan Wu**[1]  and  **Weiming Lu**[1,2,†]

[1]College of Computer Science and Technology, Zhejiang University
[2]Alibaba-Zhejiang University Joint Research Institute of Frontier Technologies
{wulinjuan525,luwm}@zju.edu.cn

## Abstract

Cross-lingual transfer learning heavily relies on well-aligned cross-lingual representations. The syntactic structure is recognized as beneficial for cross-lingual transfer, but limited researches utilize it for aligning representation in multilingual pre-trained language models (PLMs). Additionally, existing methods require syntactic labels that are difficult to obtain and of poor quality for low-resource languages. To address this gap, we propose Struct-XLM, a novel multilingual language model that leverages reinforcement learning (RL) to autonomously discover universal syntactic structures for improving the cross-lingual representation alignment of PLM. Struct-XLM integrates a policy network (PNet) and a translation ranking task. The PNet is designed to discover structural information and integrate it into the last layer of the PLM through the structural multi-head attention module to obtain structural representation. The translation ranking task obtains a delayed reward based on the structural representation to optimize the PNet while improving the alignment of cross-lingual representation. Experiments show the effectiveness of the proposed approach for enhancing cross-lingual transfer of multilingual PLM on the XTREME benchmark[1].

## 1 Introduction

Cross-lingual representation is a crucial component in the development of multilingual natural language processing (NLP) systems (Hu et al., 2020). Various multilingual Pre-trained Language Models (PLMs) like XLM-R (Conneau et al., 2020), XY-LENT (Patra et al., 2022), and VECO2.0 (Zhang et al., 2023) have demonstrated their effectiveness in cross-lingual transfer learning.

The alignment of cross-lingual representation is vital for effective cross-lingual transfer learning. Previous researches extend the pre-training tasks by introducing novel pre-training objectives at different levels of granularity, such as token-level (Luo et al., 2021; Zhang et al., 2023), word-level (Huang et al., 2019; Wei et al., 2021), and sentence-level (Huang et al., 2019; Wei et al., 2021; Chi et al., 2021; Ouyang et al., 2021). These objectives provide explicit guidance for the model to learn aligned information across languages. Some models (Luo et al., 2021; Ouyang et al., 2021) proposed cross-lingual attention mechanisms to aligned cross-lingual representation. However, there is still limited research on aligning cross-lingual representations with syntactic structure-level information.

The performance of cross-lingual transfer is also influenced by structural differences among languages (Ahmad et al., 2021; Wu et al., 2022a). And the incorporation of structural information, particularly syntax, has proven beneficial. Syntax-augmented mBERT (Ahmad et al., 2021) proposed a Graph Attention Network (GAN) to learn universal dependency tree labels and integrate them into the self-attention mechanism of mBERT for enhancing cross-lingual transfer. The utilization of syntactic structures also demonstrates improvements in various cross-lingual tasks, such as cross-lingual structured sentiment analysis (Zhang et al., 2022), cross-lingual semantic role labeling (Fei et al., 2020), and cross-lingual word sense disambiguation (Zhu, 2020). However, existing methods heavily rely on dependency treebanks or syntactic tagging tools to obtain syntactic labels, which are often limited in quantity or quality, particularly for low-resource languages.

In this paper, we propose Struct-XLM, a multilingual language model with structure discovery based on multilingual PLM, that aims to improve the alignment of cross-lingual representations of PLM by automatically learning universal structural information, such as constituent syntactic structure.

---

[†]Corresponding authors.

[1]The code is available at https://github.com/wulinjuan/Struct-XLM

As shown in Figure 1, we employ reinforcement learning (RL) to discover these syntactic structures without explicit structure annotations. Following Zhang et al. (2018), we formulate the structure discovery as a sequential decision problem optimized through policy gradient and implemented by a masked self-attention mechanism. Additionally, we introduce a structural multi-head attention mechanism, as shown in Figure 2, to effectively integrate the acquired structural information into the last layer of the multilingual pre-trained LM. Our model follows a Siamese framework and is trained using a policy network (PNet) and translation ranking task, in which the latter provides delayed rewards for RL while enhancing cross-lingual alignment. During the fine-tuning phase for downstream tasks, the parameters of the PNet are kept frozen.

To summarize, our main contributions are as follows.

- We propose Struct-XLM, a novel multilingual language model that autonomously discovers universal structural information with RL to enhance cross-lingual alignment without explicit structure annotations.

- We introduce a structural multi-head attention mechanism that integrates the learned structural information into the last layer of the multilingual PLM for improving cross-lingual transfer ability.

- Experiments on the 7 tasks of XTREME, our model shows superior average performance compared to the baseline PLM by 4.1 points and is competitive with the InfoXLM model using only 1/5000 of the amount of training data.

## 2 Related Work

### 2.1 Cross-lingual Representation Alignment

Multilingual pre-trained models such as mBERT (Pires et al., 2019), XLM (Conneau and Lample, 2019), and XLM-R (Conneau et al., 2020) have exhibited promising abilities of cross-lingual transfer, even in the absence of explicit encouragement of learning aligned representations across languages. These models are typically pre-trained with multilingual masked language modeling (MMLM) (Pires et al., 2019) and translation masked language modeling (TLM) (Conneau and Lample, 2019).

Subsequent works designed more pre-training objectives to guide representation alignment with parallel corpora at different granularities. For instance, Unicoder (Huang et al., 2019) integrates cross-lingual word discovery (CWD) and cross-lingual paraphrase classification (CPC) to learn the underlying word alignments between two languages and provide an explicit focus on the alignment of sentences. InfoXLM (Chi et al., 2021) adds a sentence-level contrastive learning loss for bilingual pairs to maximize the mutual information between translation sentence pairs. HICTL (Wei et al., 2021) proposed sentence-level and word-level contrastive learning to distinguish parallel sentences and related words. VECO2.0 (Zhang et al., 2023) adds token-level contrastive loss for synonym alignment with the thesaurus dictionary. In addition to using parallel corpora, ERNIE-M (Ouyang et al., 2021) integrates back-translation into the pre-training process to generate pseudo-parallel pairs for monolingual corpora. Moreover, ERNIE-M (Ouyang et al., 2021) and VECO (Luo et al., 2021) proposed a cross-attention module that builds interdependence between languages in model inner.

Existing work has enhanced the alignment of cross-lingual representations at multiple granularities (token-level, word-level, and sentence-level). Considering that differences in syntactic structure between languages can also affect alignment, we focused on introducing syntactic structural level information into cross-lingual representations to enhance alignment. Since syntactic tags are challenging to obtain, especially for low-resource languages, we introduce reinforcement learning methods to learn multilingual structural information without explicit structure annotations and guide the model to learn cross-lingual aligned representations on machine translation ranking tasks.

### 2.2 Structure-augmented Cross-lingual transfer

The use of universal dependency parse structures has been found to be beneficial for cross-lingual transfer in cross-lingual NLP tasks (Ahmad et al., 2021; Fei et al., 2020; Zhang et al., 2022; Zhu, 2020; Xu et al., 2022). Many approaches have utilized these structures by fusing them with input sequences or incorporating them into self-attention mechanisms to enhance the learning of language syntax. Most of the work is task-oriented(Zhang et al., 2022; Wu et al., 2020; Zhu, 2020; Wu et al.,

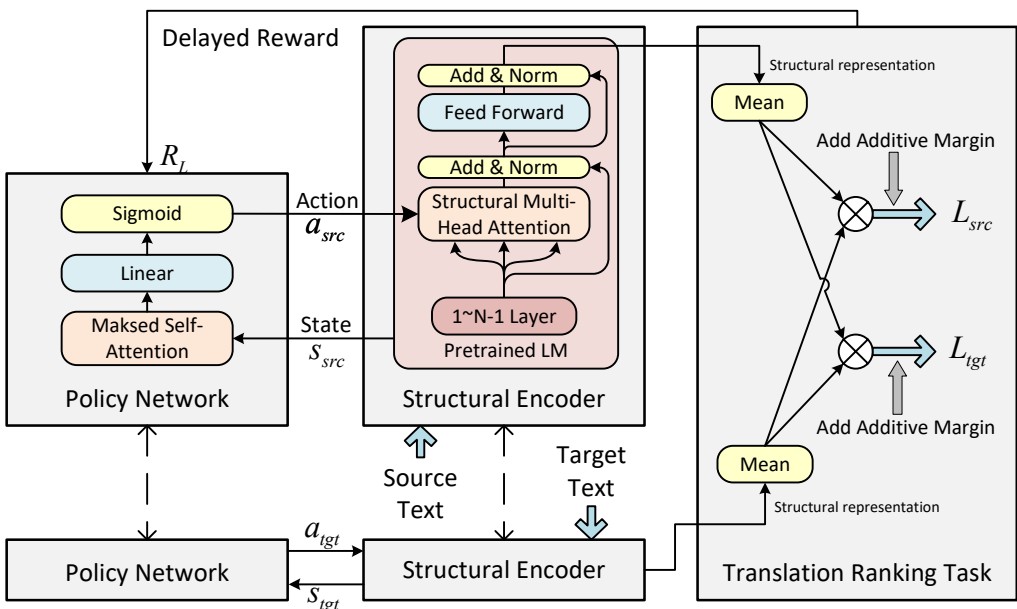

Figure 1: The framework of Struct-XLM. The policy network (PNet) samples an action at each state. The structural encoder offers state representation to PNet and outputs the final representation to calculate translation ranking loss, which provides a delayed reward to PNet.

2022b; Xu et al., 2022; Wu et al., 2023), but Syntax-augmented mBERT (Ahmad et al., 2021) enables structural learning without the inference phase by learning structural encoders and incorporating structural coding into self-attention mechanisms. So it can be generalized to arbitrary cross-lingual understanding tasks. However, these approaches require syntactically labeled data, which can be difficult to obtain in high quality across multiple languages.

To address this challenge, we follow Zhang et al. (2018) to utilize reinforcement learning to learn the universal structure for enhancing cross-lingual transfer without explicit structure annotations. Zhang et al. (2018) proposed an RL method to learn monolingual sentence representation by discovering structure for text classification. We extended their method to improve the alignment of cross-lingual representation.

## 3 Method

### 3.1 Overview

In this paper, we propose a framework that integrates structural information to enhance the cross-lingual transfer ability of multilingual PLM by improving the alignment of cross-lingual representations. The framework of the Struct-XLM model, shown in Figure 1, consists of three key components: the Policy Network (PNet), Structural Encoder, and Translation Ranking Task module. We

adopt a Siamese framework to learn representations from parallel sentence pairs $(X, Y)$, with shared parameters across components. The structural encoder is initialized with a multilingual PLM and provides state representation $S$ to PNet. PNet utilizes a stochastic policy to sample actions at each state, generating an action vector $\mathbf{a} = a^1 a^2 \cdots a^T$ for the sentence $\mathbf{x} = x^1 x^2 \cdots x^T \in X$ comprising T tokens. The action vector can represent the structure of the sentence. As depicted in Figure 2, we convert the action vector into an action matrix and feed it into the scaled dot-product self-attention module to construct a structural multi-head attention module in the last layer of PLM. The Structural Encoder then outputs a multilingual structural representation for the Translation Ranking Task module to calculate the ranking loss and provides a delayed reward to PNet. The reinforcement learning process is naturally handled using the policy gradient method (Sutton et al., 1999).

These three components are interleaved together: the state representation of PNet is derived from the Structural Encoder, the calculation of ranking loss relies on the final structured representation, and PNet receives rewards from the Translation Ranking Task module to guide policy learning.

### 3.2 Structural Encoder

Our structural encoder is initialized by multilingual PLM. For learning structural representation, we propose a structural multi-head attention module

in the last layer of PLM, and one of the structural self-attention flowing is shown in Figure 2. The original attention probability matrix is denoted as:

$$E = \text{softmax}(\frac{QK^T}{\sqrt{d_k}}) \quad (1)$$

where the dot-product is scaled by $\frac{1}{\sqrt{d_k}}$, $d_k$ denotes the dimension of query matrix $Q$ and key matrix $K$. In our structural multi-head attention module, the $E$ is not only determined by $Q$ and $K$ but also Action Matrix $A$ generated from the actor vector, which guides each word only pay attention to the words within the same constituent.

A sentence consists of constituents, such as noun phrase or verb phrase. The action vector from PNet provides information about the boundaries of constituent structure in sentences ('0' and '1' denote the word is inside or is a end boundary of a constituent), where split the sentence into several constituents. In Figure 2, for example, the phrase '*a competitive edge*' is identified as a constituent $c$ by the sub action vector $\{0, 0, 1\}$. This action vector is then converted into a sub-action matrix:

$$A^c = \begin{bmatrix} 0 & 1 & 1 \\ 1 & 0 & 1 \\ 1 & 1 & 0 \end{bmatrix} \quad (2)$$

where $A^c_{ij} = 1$ represents $i$th word pay attention to the $j$th word in this constituent. There is a special case when only one word is divided into a single constituent, its sub-action matrix is an identity matrix. The final Action Matrix $A$ is obtained by splicing all the sub-action matrices by the diagonal and filling with 0 on the remaining positions. The detail of algorithm can be see in Appendix B

By combining the Action Matrix $A$ with the softmax attention mechanism, we calculate the structural attention probability $E$ as follows:

$$E = A \odot \text{softmax}(\frac{QK^T}{\sqrt{d_k}}) \quad (3)$$

where $\odot$ represents element-wise multiplication. In the case of multi-head attention, $n$ different heads share the same action matrix $A$ and obtain the structural representation $h$ with the dimension of $d_{model} = n \times d_k$. The structural representation is then transformed into a sentence embedding using mean-pooling. Finally, the translation ranking loss is calculated based on this sentence embedding.

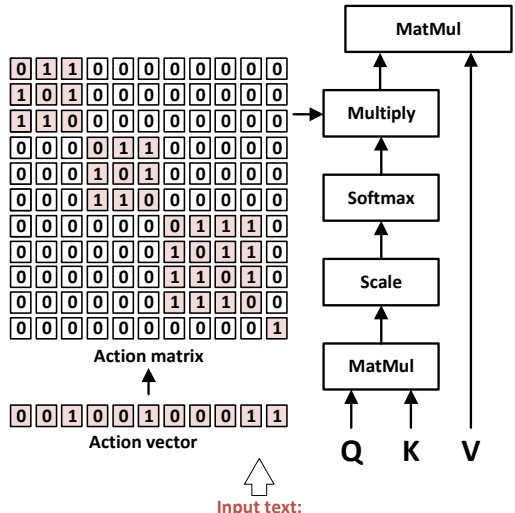

Figure 2: The framework of structural self-attention.

### 3.3 Translation Ranking Task

For enhancing cross-lingual representation, we introduce the translation ranking task with in-batch negative sampling(Yang et al., 2019a; Feng et al., 2022):

$$\mathcal{L} = -\frac{1}{N}\sum_{i=1}^{N} log \frac{e^{\phi(x_i, y_j) - m}}{e^{\phi(x_i, y_j) - m} + \sum_{\substack{n=1, \\ n \neq j}}^{N} e^{\phi(x_i, y_n)}} \quad (4)$$

The embedding space similarity of sentence pairs $x$ and $y$ is given by $\phi(x, y) = \text{mean}(h_x)\text{mean}(h_y)^T$. $m$ is an additive margin around positive pairs, which improves the separation between translations and non-translations.

Considering that there are n-way parallel pairs in data, we introduce in-batch labels $\mathbf{l} \in \mathbb{R}^{N \times N}$ to improve the translation ranking loss. $\mathbf{l}_{ij} = 1$ represents $y_j$ and $x_i$ are translation pair and $\mathbf{l}_{ij} = 0$ represents $y_j$ and $x_i$ are non-translation pair. The $N$ is batch size. The new loss can be defined as:

$$\mathcal{L} = -\frac{1}{N}\sum_{i=1}^{N} log \frac{\sum_{j=1, \mathbf{l}_{ij}=1}^{N} e^{\phi(x_i, y_j) - m}}{\sum_{\substack{j=1, \\ \mathbf{l}_{ij}=1}}^{N} e^{\phi(x_i, y_j) - m} + \sum_{\substack{n=1, \\ \mathbf{l}_{in}=0}}^{N} e^{\phi(x_i, y_n)}} \quad (5)$$

The loss aims to rank the true translations of $x_i$ over all other alternatives in the same batch. $\mathcal{L}$ is asymmetric and depends on whether the softmax is applied to the source or target sentences. To achieve bidirectional symmetry, the final loss can

be obtained by summing the source-to-target loss, $\mathcal{L}$, and the target-to-source loss, $\mathcal{L}'$:

$$\mathcal{L}_{trans} = \mathcal{L} + \mathcal{L}' \qquad (6)$$

### 3.4 Policy Network

The policy network includes a masked self-attention module, linear layer, and sigmoid layer. The masked self-attention limited each state $s_t$ only to notice the previous state and predict the action in the sequence. We briefly introduce state, action and policy, reward, and objective function as follows:

**State** Each state vector $s_t$ with a dimension of $d_{model}$ is each token representation from the Structural Encoder without the input of the action vector. The complete state representation $S = (s_1, s_2, \ldots, s_T) \in \mathbb{R}^{T \times d_{model}}$ is encoded into an action embedding space using a masked self-attention layer and a linear layer.

**Action and Policy** We adopt binary actions {*Inside*, *End*} to discover constituent structure, indicating that a word is inside or at the end of a constituent. PNet adopts a stochastic policy $\pi(a_t|s_t; \Theta)$, which represents the probability of selecting action $a_t$ at state $t$. The policy is defined as follows:

$$\pi(a|S; \Theta) = \sigma(\mathbf{W} * \mathrm{SA}(S) + \mathbf{b}) \qquad (7)$$

$$\mathrm{SA}(S) = \mathrm{softmax}(\frac{QK^T}{\sqrt{d_{model}}}) \cdot V, \\ Q = W^Q S, K = W^K S, V = W^V S \qquad (8)$$

where $\sigma$ represents the sigmoid function, $\Theta = \{(W^Q, W^k, W^V), W, b\}$ denotes the parameters of PNet. $\mathbf{W}^Q$, $\mathbf{W}^K$, and $\mathbf{W}^V$ are the weights for the query matrix $Q$, key matrix $K$, and value matrix $V$ of masked self-attention layer $\mathrm{SA}(\cdot)$, with dimension $d_{model}$. The $\mathbf{W}$ and $\mathbf{b}$ are the weight and bias of the linear layer. During training, the action is sampled according to the probability in Eq. 7. During the test, the action with the maximal probability (i.e., $a_t^* = argmax_a \pi(a_t|s_t; \Theta)$) is chosen to obtain the action vector prediction.

**Reward** After sampling all the actions from PNet, the structural representation of a sentence $x$ is determined by our Structural Encoder. This representation is then used to calculate the translation

ranking loss and reward. The reward is defined as:

$$R_L = log \frac{\sum\limits_{j=1, \mathbf{l}_j=1}^{N} e^{\phi(x, y_j) - m}}{\sum\limits_{j=1, \mathbf{l}_j=1}^{N} e^{\phi(x, y_j) - m} + \sum\limits_{n=1, \mathbf{l}_n=0}^{N} e^{\phi(x, y_n)}} \qquad (9)$$

where the $\mathbf{l}_j$ is the label indicates the $y_j$ whether a translation of $x$. This reward is considered a delayed reward since it is obtained after constructing the entire representation.

**Objective Function** Following Zhang et al. (2018), we optimize the parameters of PNet using REINFORCE algorithm (Sutton et al., 1999) and policy gradient methods, aiming to maximize the expected reward as shown below.

$$\mathcal{J}(\Theta) = \sum_{s_1 a_1 \cdots s_T a_T} \prod_t \pi_\Theta(a_t|s_t) R_L \qquad (10)$$

and the gradient with likelihood ratio trick is defined as follows:

$$\triangledown_\Theta \mathcal{J}(\Theta) = \sum_{t=1}^{T} R_L \triangledown_\Theta \log \pi_\Theta(a_t|s_t) \qquad (11)$$

### 3.5 Training Process

Refer to Zhang et al. (2018), our training process consists of three steps. Firstly, we pre-train the Structural Encoder using the Translation Ranking Task. Then, we pre-train the PNet while keeping the parameters of the other two models fixed. Finally, we jointly train all three components. To alleviate the challenges of training RL from scratch and reduce variance, we also adopt a warm-start approach in the first step. Specifically, we collect a corpus of 13.7k sentence pairs from the UD 2.9 Treebank (Zeman et al., 2021), which provides syntax tree labels. These labels are only used as a heuristic signal to split sentences into constituents solely for the warm start in the first step.

## 4 Experiments

### 4.1 Implementation Details

We used the XLM-R$_{large}$ model as a base pre-trained LM to initialize the Structural Encoder, and the whole Struct-XLM model has 564M parameters (559M for fine-tuning). To train our model, we collected an 8MB parallel corpus, including 13.7k sentence pairs (English is the source language), and more detail in Appendix A. In order to facilitate

| Datasets | Pair sentence | | Structured prediction | | Question answering | | | | | | AVG |
|---|---|---|---|---|---|---|---|---|---|---|---|
| | XNLI | PAWS-X | POS | NER | XQuAD | | MLQA | | TyDiQA | | |
| #Langs | 15 | 7 | 33 | 40 | 11 | | 7 | | 9 | | |
| Metrics | Acc. | Acc. | F1 | F1 | F1 | EM | F1 | EM | F1 | EM | |
| Model | | | | | | | | | | | |
| XLM-R$_{large}$ (Hu et al., 2020) | 79.2 | 86.4 | 73.8 | 65.4 | 76.6 | 60.8 | 71.6 | 53.2 | 65.1 | 45.0 | 67.7 |
| InfoXLM(Chi et al., 2021) | 81.4 | – | – | – | – | – | 73.6 | 55.2 | – | – | – |
| ERNIE-M(Ouyang et al., 2021) | 81.9 | 89.5 | – | – | – | – | **73.7** | **55.3** | – | – | – |
| XY-LENT(Patra et al., 2022) | 80.5 | 89.7 | – | – | 76.8 | 62.1 | 71.3 | 53.2 | 67.1 | 51.5 | – |
| VECO2.0(Zhang et al., 2023) | 80.3 | 88.5 | 75.4 | 67.2 | 78.9 | 63.7 | 72.7 | 54.2 | 71.1 | 54.7 | 70.7 |
| ERNIE-M$^\dagger$ | 80.1 | 87.9 | 76.7 | 65.9 | 78.2 | 62.7 | 72.8 | 54.6 | 70.3 | 52.7 | 70.2 |
| InfoXLM$^\dagger$ | **82.1** | 89.3 | 75.6 | 67.8 | 79.3 | 63.8 | 73.5 | 55.1 | 72.2 | 54.6 | 71.3 |
| Struct-XLM | 81.2 | **90.1** | **77.6** | **68.6** | **79.5** | **64.1** | 73.2 | 54.8 | **73.1** | **56.0** | **71.8** |

Table 1: Results of 7 tasks on XTREME benchmark. The detailed results for each language are in Appendix D. † denotes the results from our re-implement.

the smooth update of the policy gradient during training, we introduced a suppression factor that is multiplied to Eq. 11. This factor is set to 0.5. During the training process, we employed the Adam optimization algorithm to update the model parameters. The initial learning rate was set to 6e-6 for the first step, which ran for 5 epochs. The second step ran for 2 epochs with a learning rate of 3e-6, and the last step utilized 5 epochs with a learning rate of 3e-6. The batch size is 5 for all steps.

## 4.2 Baselines

Except for the base model XLM-R$_{large}$, we compare the cross-lingual performance of our proposed model against 4 strong cross-lingual models:

- **Info-XLM** focuses on maximizing mutual information between translation pairs. We compare Struct-XLM with its large size version (558M parameter), which continues training XLM-R$_{large}$ on 42GB parallel corpora.
- **ERNIE-M** incorporates cross-attention masked language modeling on both parallel and monolingual corpora. The parameter size of the large version model and the training data is the same as Info-XLM.
- **XY-LENT** leverages novel sampling strategies with X-Y bitexts to learn the cross-lingual alignment representation. It has a twice bigger vocabulary size than XLM-R$_{large}$ but with a smaller parameter size (477M for the base version) by training on ELECTRA-style tasks.
- **VECO2.0** bridges the representations of synonym pairs embedded in the bilingual corpus based on VECO. It has 559M parameters and is trained with 2.5TB monolingual data and 4TB parallel pairs.

We do not compare our model with Syntax-augmented mBERT due to its small parameter size.

## 4.3 Evaluation

In our evaluation, we perform experiments on seven multilingual tasks from the XTREME (Hu et al., 2020) benchmark. These tasks cover a wide range of languages. For sentence-pair classification, we evaluate on the Cross-lingual Natrual Language Inference dataset (XNLI) (Conneau et al., 2018), and Cross-lingual Paraphrase Adversaries from Word Scrambling dataset (PAWS-X) (Yang et al., 2019b). For structured prediction, POS tagging from the Universal Dependencies v2.5 treebanks (Nvire et al., 2020) and NER from Wikiann (Pan et al., 2017). For cross-lingual question answering: XQuAD (Artetxe et al., 2020), MLQA (Lewis et al., 2020), and the gold passage version of the TyDiQA dataset (TyDiQA-GoldP) (Clark et al., 2020). Here, for a cross-lingual setting, all tasks provide English training data and dev/test set in all involved languages. We consider the zero-shot cross-lingual setting, i.e. only using the English data for fine-tuning. The hyper-parameters setting for fine-tuning are shown in Appendix C.

## 4.4 Results

In Table 1, we present the results of Struct-XLM and baselines on the seven tasks of XTREME. The results demonstrate that Struct-XLM outperforms XLM-R$_{large}$ and latest model VECO2.0 (Zhang et al., 2023) on all tasks, with an average improvement of 4.1% and 1.1% respectively. Compared to the best strong baseline InfoXLM (Chi et al., 2021), we have a competitive average performance and we only use 8MB of training data, which is

| Datasets | Pair sentence | | Structured prediction | | | | Question answering | | | | AVG |
|---|---|---|---|---|---|---|---|---|---|---|---|
| | XNLI | PAWS-X | POS | NER | XQuAD(F1) | XQuAD(EM) | MLQA(F1) | MLQA(EM) | TyDiQA(F1) | TyDiQA(EM) | |
| XLM-R$_{large}$(Hu et al., 2020) | 10.1 | 9.7 | 23.0 | 19.8 | 10.9 | 16.4 | 13.9 | 20.3 | 7.2 | 13.3 | 14.5 |
| InfoXLM(Chi et al., 2021) | 8.9 | – | – | – | – | – | 12.7 | 19.1 | – | – | – |
| ERNIE-M(Ouyang et al., 2021) | 7.9 | 7.5 | – | – | – | – | **12.5** | **18.9** | – | – | – |
| XY-LENT(Patra et al., 2022) | **7.8** | 6.8 | – | – | 11.4 | 15.3 | 13.8 | 20.0 | 7.1 | 8.6 | – |
| VECO2.0 | 9.2 | 8.6 | 21.4 | 18.1 | 10.2 | 16.2 | 13.3 | 20.0 | **2.2** | **6.8** | 12.6 |
| ERNIE-M$^†$ | 8.8 | 9.3 | 20.0 | 18.4 | 10.1 | **15.0** | 12.8 | 19.0 | 4.0 | 7.0 | 12.4 |
| InfoXLM$^†$ | 8.5 | **6.7** | 21.0 | 17.2 | 10.3 | 16.0 | 13.1 | 19.4 | 4.7 | 10.5 | 12.7 |
| Struct-XLM | 9.7 | 6.9 | **19.0** | **15.8** | **9.6** | 15.6 | 13.1 | 19.6 | 2.9 | 9.6 | **12.2** |

Table 2: The transfer gap of different models on XTREME tasks, where the lower the score the better transferability.

| Datasets | Pair sentence | | Structured prediction | | Question answering | | | | | | AVG |
|---|---|---|---|---|---|---|---|---|---|---|---|
| | XNLI | PAWS-X | POS | NER | XQuAD | | MLQA | | TyDiQA | | |
| Metrics | Acc. | Acc. | F1 | F1 | F1 | EM | F1 | EM | F1 | EM | |
| XLM-R$_{large}$(Hu et al., 2020) | 79.2 | 86.4 | 73.8 | 65.4 | 76.6 | 60.8 | 71.6 | 53.2 | 65.1 | 45.0 | 67.7 |
| Struct-XLM | **81.2** | **90.1** | **77.6** | **68.6** | **79.5** | **64.1** | **73.2** | **54.8** | **73.1** | **56.0** | **71.8** |
| w/o action | 80.4 | 89.4 | 76.1 | 66.7 | 78.9 | 63.5 | 72.6 | 54.2 | 71.6 | 54.1 | 70.7 |
| w/o warm-start | 79.7 | 89.0 | 76.3 | 65.6 | 78.6 | 62.8 | 72.0 | 54.0 | 71.8 | 53.7 | 70.4 |
| only warm-start | 79.7 | 88.5 | 75.0 | 64.5 | 78.1 | 62.4 | 71.7 | 53.8 | 70.8 | 52.8 | 69.7 |
| w/o PNet | 79.4 | 88.3 | 74.9 | 62.9 | 77.5 | 61.8 | 71.3 | 53.3 | 70.9 | 52.8 | 69.3 |

Table 3: Ablation results on each model. The detailed results for each language are in Appendix D.

| Model | de | fr | ru | zh | avg |
|---|---|---|---|---|---|
| XLM-R$_{large}$ | 67.5 | 66.5 | 73.5 | 56.7 | 66.0 |
| VECO2.0 | 90.5 | 86.1 | 88.1 | 80.2 | 86.2 |
| XLM-R$_{large}^†$ | 82.9 | 72.5 | 79.8 | 66.1 | 75.3 |
| InfoXLM$^†$ | 92.7 | 89.1 | **92.2** | 81.6 | 88.9 |
| Struct-XLM | **93.5** | **90.4** | 92.2 | **84.4** | **90.1** |

Table 4: BUCC F1 scores for each language. † denotes the results from our re-implementation.

1/5000 of parallel data used by InfoXLM. Struct-XLM also surpasses XY-LENT on 8/10 metrics, with an average advancement of 2.5%.

In sentence-pair classification tasks, Struct-XLM performs slightly better than the best model XY-LENT on the PAWS-X by 0.4%. However, on the XNLI, it is inferior to Info-XLM by 0.9%. This may be because the translation ranking task in Struct-XLM focuses more on detecting meaning equivalence rather than fine-grained distinctions of meaning overlap. For structured prediction tasks, Struct-XLM outperforms all baselines and has the lowest transfer gap as shown in Table 2, which can be attributed to the structure discovery. In question answering, Struct-XLM is inferior to the best EM score by 0.5% on MLQA. However, it achieves the best performance on the XQuAD and TyDiQA datasets. It surpasses InfoXLM by 0.9% and 1.4% in EM and F1 metrics on TyDiQA, respectively.

Overall, Struct-XLM delivers impressive performance with only an 8MB parallel corpus. The improvement can be attributed to the multilingual structured representation alignment achieved through RL and the translation ranking task. Further investigation and analysis will be conducted in the ablation study and analysis experiments.

# 5 Analysis

## 5.1 Transfer Gap

Table 2 presents the transfer gap of Struct-XLM and baselines on the XTREME benchmark. The transfer gap refers to the difference in performance

between the English test set and the average performance across other languages. It is observed that the transfer gap of Struct-XLM is relatively high for certain metrics, such as accuracy in XNLI and PAWS-X, and EM in TyDiQA. This is because the model's performance in English has significantly improved. However, Struct-XLM demonstrates a lower average transfer gap than baselines, indicating improved cross-lingual transferability.

## 5.2 Ablation study

To explore the impact of PNet and RL, we fine-tune the Structural Encoder Part (i.e. w/o action) or the model only trained by the first step (i.e. only warm-start). To discuss the significance of the warm-start step, we also evaluate the ablation model without it (w/o warm-start). Moreover, we train a language model that only has Structural Encoder and Translation Ranking Task module (i.e. w/o PNet). The results are shown in Table 3, and the results of Struct-XLM in 7 tasks always be the best. Specifically, the Struct-XLM only trained on the first

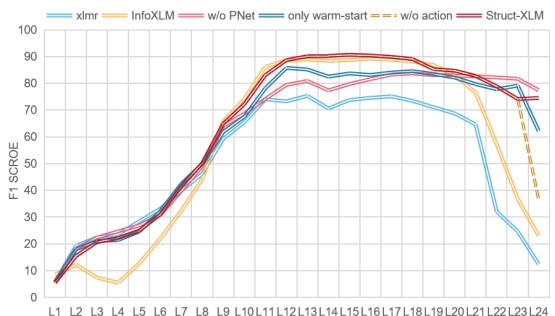

Figure 3: Evaluation results (F1 score) of different layers on BUCC cross-lingual sentence retrieval.

step (i.e. only warm-start) surpasses the language model without PNet (i.e. w/o PNet) with an average advancement of 0.4%, which indicates the significance of the simple structural inspire information to improve the cross-lingual transfer ability. The evaluation of the model without action (i.e. w/o action) also obtains a greater improvement than baseline XLM-R$_{large}$ by 3.0% on the average. This suggests that RL training greatly enhances the cross-lingual transfer ability of the multilingual PLM, even without the subsequent use of the PNet module in fine-tuning. The model w/o warm-start doesn't surpass the Struct-XLM model, confirming the effectiveness of our warm-start procedure. From the results, we can observe that all components and training settings contribute to overall performance improvement.

## 5.3 Cross-Lingual Representations

To assess the alignment of cross-lingual sentence representations obtained from our Struct-XLM, we introduce the BUCC sentence retrieval task. Table 4 presents the F1 scores of four languages on BUCC, where the representations of our model and XLM-R$_{large}$ and InfoXLM are extracted from the middle layer (13th layer) following the choice in Hu et al. (2020). The evaluation results demonstrate that Struct-XLM produces better-aligned cross-lingual sentence representations compared to XLM-R$_{large}$ and InfoXLM, achieving an average performance improvement of 14.8% and 1.2%, respectively.

Furthermore, we analyze the average results of all layers of the models on BUCC, as depicted in Figure 3. For XLM-R$_{large}$ and InfoXLM, we observe a performance drop in the last few layers, which is expected as these layers are encouraged to focus on token-level embedding due to the pre-trained objectives. On the other hand, Struct-XLM

consistently achieves high retrieval F1 scores even in the last few layers. Although the last layer of Struct-XLM without the action component (i.e., w/o action) exhibits a lower F1 score, it still benefits from enhancing the aligned knowledge of other layers to achieve excellent performance on XTREME.

Structural probe (Hewitt and Manning, 2019; Chi et al., 2020) demonstrates that the middle layers of PLMs capture richer syntactic information. Struct-XLM obtain significant improvements in the middle layers, indicating that it has learned structural information and provides better-aligned representations compared to XLM-R$_{large}$.

| Model | $P_{en}$ | $R_{en}$ | $F1_{en}$ | $P_{tgt}$ | $R_{tgt}$ | $F1_{tgt}$ |
|---|---|---|---|---|---|---|
| Struct-XLM | **34.49** | **66.84** | **45.50** | **35.67** | **57.58** | **43.14** |
| w/o warm-start | 30.57 | 62.13 | 40.98 | 31.63 | 56.48 | 39.67 |

Table 5: The precision(P), recall(R), and F1 scores of the constituent boundary prediction.

## 5.4 Analyze of Discovered Structure

**Quantitatively Analyze** Struct-XLM aim to acquire structural information that facilitates cross-lingual alignment, we employ retrieval tasks in Section 5.3 to quantitatively analyze whether the information obtained through RL contributes to cross-lingual alignment. Furthermore, we try to conduct a quantitative analysis of the discovered structure by evaluating the constituents' boundary predicted by the action vector. The labeled test set is from the warm-start step, including English (en) and target languages (tgt) two-way. where the recall of the boundaries predicted by Struct-XLM is 66.84% and 57.58% in English and target language, respectively, which suggests that Struct-XLM is able to correctly predict the constituent boundaries in the labeled test set to some extent. Perhaps Struct-XLM tends to predict more constituents, so it gets low precision (P). We also report the results of the model w/o warm-start to prove the influence of the warm-start step for structural discovery.

**Qualitatively Analyze** Table 6 presents some interesting structures discovered by Struct-XLM. Some constituents can form a sentence with complete meaning, such as "乘客乘坐缆车" (Passengers ride the gondola), while others may consist of just a single word. In these examples, the structures discovered by Struct-XLM exhibit more flexibility in terms of length and amount of constituents compared to predefined structures. In the case of En-

| Lang. | Structure | Sentence |
|---|---|---|
| en | Predefined | The feasibility study │ estimates that it would take passengers │ about four │ minutes │ to cross the Potomac River │ on the gondola. |
|  | from Struct-XLM | The feasibility study estimates that it would take passengers about four minutes │ to cross the Potomac River │ on the gondola. |
| fi | Predefined | Toteutettavuustutkimuksessa arvioidaan, │ että matkustajilta kuluisi │ noin neljä │ minuuttia │ Potomac River -joen │ ylittämiseen korihississillä. |
|  | from Struct-XLM | Toteutettavuustutkimuksessa arvioidaan, että matkustajilta kuluisi │ noin neljä minuuttia │ Potomac │ River -joen │ ylittämiseen │ korihississillä. |
| de | Predefined | Der Machbarkeitsstudie zufolge │ könnten die Passagiere den Potomac River mit der Gondel in ungefähr 4 Minuten überqueren. |
|  | from Struct-XLM | Der │ Machbarkeitsstudie │ zufolge │ könnten die Passagiere den Potomac River │ mit der Gondel in ungefähr 4 Minuten überqueren. |
| pt | Predefined | O estudo de viabilidade │ estima que os passageiros iriam levar quatro minutos │ para atravessar o Rio Potomac │ em a gôndola. |
|  | from Struct-XLM | O estudo de viabilidade │ estima que os │ passageiros │ iriam │ levar quatro minutos para atravessar o │ Rio Potomac em a gôndola. |
| pl | Predefined | Na podstawie wyliczeń │ szacuje się, │ że przekroczenie rzeki Potomac gondolą zajęłoby pasażerem │ około │ czterech minut. |
|  | from Struct-XLM | Na podstawie wyliczeń │ szacuje się, że przekroczenie rzeki │ Potomac gondolą zajęłoby │ pasażerom około │ czterech minut. |
| zh | Predefined | 根据可行性研究估计， │ 乘客乘坐缆车 穿越波托马克河 │ 需时约四分钟。 |
|  | from Struct-XLM | 根据 │ 可行性研究 │ 估计， │ 乘客乘坐缆车 │ 穿越波托马克河 │ 需时约 │ 四分钟。 |

Table 6: The comparison of the predefined structures and those discovered by Struct-XLM in multiple languages.

glish, being a source language aligned with multiple languages, the number of constituents obtained tends to be smaller. However, Struct-XLM sometimes struggles to identify the correct boundaries between phrases, as seen in the example where the term "Potomac River" in Finnish (fi) is split into two parts. This task is extremely challenging for any model without explicit structure annotations.

## 6 Conclusion

In conclusion, we introduced Struct-XLM, a novel approach to enhance cross-lingual transfer learning in multilingual language models. By leveraging reinforcement learning without explicit structure annotations, the Struct-XLM discovers structures and improves the alignment of cross-lingual representations. Specifically, we employ policy gradient RL and a translation ranking task to discover constituent syntactic structures and enhance cross-lingual transfer. Experimental results on seven XTREME benchmark tasks demonstrated the effectiveness of Struct-XLM, outperforming baseline PLM models with an average improvement of 4.1 points and being competitive with the best strong baseline with small training data. Our analysis also revealed that Struct-XLM improves performance on the sentence retrieval task BUCC, particularly in the middle layers of the pre-trained model, indicating the acquisition of valuable structural information for better-aligned representations.

## Acknowledgement

This work is supported by the Fundamental Research Funds for the Central Universities (No. 226-2023-00060), the Key Research and Development Program of Zhejiang Province (No. 2021C01013), National Key Research and Development Project of China (No. 2018AAA0101900), and MOE Engineering Research Center of Digital Library.

## Limitations

While Struct-XLM presents advancements in improving cross-lingual representation alignment, it also has some limitations. We outline these limitations in this section.

### Training Data

The strong baseline models we compare against may benefit from large amounts of parallel data for training, which contributes to their superior performance. Since policy gradient RL learning requires multiple sampling learning for each sample, our model takes about 30 hours to complete three stages of training on an NVIDIA RTX A6000 using 8MB of training data. In our case, due to computational constraints, we were only able to utilize a limited amount of parallel data (8M) for training. Increasing the amount of parallel data without labels in the last two steps could potentially improve the performance of our model and provide a better-aligned cross-lingual representation.

### Learning Structure from Middle Layer

As mentioned, the structural probe (Chi et al., 2020) has shown that the middle layer of multilingual PLMs captures rich syntactic information. However, it is worth considering that syntax and semantics are intertwined, and the primary goal of cross-lingual alignment is semantic alignment. Therefore, we explore whether structures learned from the final layer, which captures more semantic information, can enhance alignment. If structures discovered from the middle layer representations can further improve alignment, it would provide additional evidence for the effectiveness of our method.

### Siamese Framework vs. Single Tower Version

The Siamese framework used in Struct-XLM, which employs separate encoders for parallel sen-

tence pairs, is effective in capturing cross-lingual alignment (Guo et al., 2018; Yang et al., 2019a; Feng et al., 2022). However, its token-level alignment may not be sufficient. An alternative approach could involve concatenating the parallel sentence pairs into a single input, allowing for more comprehensive learning from the parallel corpus and potentially encouraging the policy network to capture more universal structures through attention mechanisms.

Overall, these limitations provide opportunities for future research to further enhance the performance and capabilities of cross-lingual representation learning models like Struct-XLM.

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

# Appendices

| Lang. | Train | Dev | Test |
|---|---|---|---|
| English(en) | 12038 | 882 | 800 |
| Swedish(sv) | 1634 | 406 | 348 |
| Italian(it) | 1528 | 76 | 109 |
| French(fr) | 951 | 45 | 83 |
| Turkish(tr) | 686 | 55 | 40 |
| German(de) | 502 | 20 | 20 |
| Chinese(zh) | 502 | 20 | 20 |
| Japanese(ja) | 502 | 20 | 20 |
| Finnish(fi) | 502 | 20 | 20 |
| Arabic(ar) | 502 | 20 | 20 |
| Spanish(es) | 502 | 20 | 20 |
| Indonesian(id) | 502 | 20 | 20 |
| Korean(ko) | 502 | 20 | 20 |
| Thai(th) | 502 | 20 | 20 |
| Czech(cs) | 502 | 20 | 20 |
| Russian(ru) | 502 | 20 | 20 |
| Hindi(hi) | 502 | 20 | 20 |
| Polish(pl) | 502 | 20 | 20 |
| Portuguese(pt) | 502 | 20 | 20 |
| Icelandic(is) | 211 | 20 | 20 |

Table 7: The statistics of training data in each language.

## A  Statistic of Training Data

The training data for Struct-XLM is sourced from various treebanks in UD 2.9 (Zeman et al., 2021). In particular, we select 20 languages (including English) parallel universal dependencies treebanks from PUD[2], Atis[3], LinES (Ahrenberg, 2007), and ParTUT[4] treebanks. English is the source language. The statistics on the number of sentences available for each language are shown in the table 7.

## B  Algorithm

The algorithm 1 is to convert the action vector to the action matrix, as mentioned in Section 3.2.

## C  Hyper-parameters for Fine-Tuning

In Table 8, we report the hyper-parameters for fine-tuning Struct-XLM on the XTREME seven tasks.

---

**Algorithm 1** Convert the action vector to the action matrix

---

**Input:** action vector $a$, length of sentence $L$
**Output:** action matrix $\mathbf{A}$
1: number of constituents $p = 1$
2: the list of constituents $C$, in which $i$th constituents $c_i$ include start index $c_i^{start}$ and end index $c_i^{end}$
3: $c_p^{start} = 0$
4: **for** $j = 1$ to $L$ **do**
5:     **if** $a[j-1] == 1$ **then**
6:         $c_p^{end} = j - 1$
7:         $p+ = 1$
8:         **if** $j < L$ **then**
9:             $c_p^{start} = j$
10:         **end if**
11:     **end if**
12: **end for**
13: $c_p^{end} = L - 1$
14: **for** $c_i$ in $C$ **do**
15:     **if** $c_i^{start} == c_i^{end}$ **then**
16:         sub-action matrix $\mathbf{A}_{c_i} = [1]$
17:     **else**
18:         $\mathbf{A}_{c_i} = \mathbf{1} - \mathbf{I}$, where $\mathbf{1}, \mathbf{I} \in \mathcal{R}^{n \times n}$, and $n = c_i^{end} - c_i^{start} + 1$ is the length of constituent $c_i$.
19:     **end if** sentence-wise sub-action matrix $\hat{\mathbf{A}}_{c_i} = [\mathbf{0}^l \mathbf{A}_{c_i} \mathbf{0}^r]$, where $\mathbf{0}^l \in \mathcal{R}^{n \times c_i^{start}}$ and $\mathbf{0}^r \in \mathcal{R}^{n \times (L - c_i^{end} - 1)}$.
20: **end for**
21: **return**

$$\mathbf{A} = \begin{bmatrix} \hat{\mathbf{A}}_{c_1} \\ \hat{\mathbf{A}}_{c_2} \\ \vdots \\ \hat{\mathbf{A}}_{c_p} \end{bmatrix}$$

---

[2] http://universaldependencies.org/conll17/
[3] https://github.com/howl-anderson/ATIS_dataset/
[4] https://github.com/msang/partut-repo

## D Results for each task and language

We show the detailed results for all tasks and languages in Tables 9 (XNLI), 10 (PAWS-X), 11 (POS), 12 (NER), 13 (XQuAD), 14 (MLQA), 15 (TyDiQA-GoldP). † denotes the results from our re-implement.

Based on the experimental results in Table 9-15, even languages without warm-start training data can show improvements. For instance, in Table 11, Basque(eu), Tagalog(tl), and Yoruba(yo) do not have training corpora, yet they exhibit significant improvements compared to the best baseline result, with increases of 1.2%, 1.9%, and 19.2%, respectively. Moreover, in Table 12, Persian(fa) and Malay(ms) without warm-start training corpora improve by 3.5% and 2.7% than the best baseline result.

| | XNLI | PAWS-X | NER | POS | XQuAD/MLQA | TyDiQA |
|---|---|---|---|---|---|---|
| Batch size | 32 | {16,32} | {16,32} | {8,16,32} | {16,32} | {16,32} |
| Learning rate | {6,8}e-6 | {8,9,10,20}e-6 | {6,···,9}e-6 | {6,···,9}e-6 | {6,8,10,20}e-6 | {9,10,20}e-6 |
| Warmup | {12500,5000} steps | {5%,10%} | {5%,10%,1%} | {5%,10%} | 10% | 10% |
| Epochs | 10 | {5,10} | {5,10} | {5,10} | {2,3,4} | {5,10,20} |

Table 8: Hyper-parameters used for fine-tuning on the downstream tasks.

| Lang. | en | ar | bg | de | el | es | fr | hi | ru | sw | th | tr | ur | vi | zh | AVG |
|---|---|---|---|---|---|---|---|---|---|---|---|---|---|---|---|---|
| XLM-R$_{large}$(Hu et al., 2020) | 88.7 | 77.2 | 83 | 82.5 | 80.8 | 83.7 | 82.2 | 75.6 | 79.1 | 71.2 | 77.4 | 78 | 71.7 | 79.3 | 78.2 | 79.2 |
| InfoXLM(Chi et al., 2021) | 89.7 | **84.5** | **85.5** | 84.1 | 83.4 | 84.2 | 81.3 | **80.9** | 80.4 | **80.8** | 78.9 | 80.9 | **77.9** | 74.8 | 73.7 | 81.4 |
| InfoXLM† | 90.1 | 80.2 | 84.5 | **86.3** | 83.4 | **85.8** | **85.7** | 78.9 | 81.1 | 75.6 | 79.0 | 81.1 | 74.8 | **82.7** | **82.7** | **82.1** |
| ERNIE-M(Ouyang et al., 2021) | 89.3 | 81.2 | 84.5 | 84.4 | **83.7** | 85.7 | 85.1 | 78.6 | **82.0** | 76.2 | **79.2** | **81.2** | 75.4 | 81.9 | 80.5 | 81.9 |
| ERNIE-M† | 88.3 | 79.1 | 83.1 | 82.8 | 82.4 | 84.0 | 82.7 | 77.2 | 80.4 | 74.2 | 76.8 | 78.4 | 72.4 | 80.6 | 78.8 | 80.1 |
| XY-LENT(Patra et al., 2022) | 87.7 | 79.7 | 83.0 | 83.7 | 82.0 | 84.7 | 83.7 | 76.1 | 81.5 | 75.5 | 77.9 | 79.3 | 71.6 | 80.3 | 80.2 | 80.5 |
| VECO2.0(Zhang et al., 2023) | 88.9 | 79.1 | 83.4 | 83.0 | 82.7 | 84.9 | 83.2 | 76.7 | 80.7 | 71.9 | 77.8 | 79.5 | 72.8 | 80.6 | 79.9 | 80.3 |
| Struct-XLM | **90.2** | 78.9 | 83.8 | 84.7 | 82.8 | 85.2 | 84.3 | 77.6 | 80.1 | 73.9 | 79.0 | 80.2 | 74.8 | 81.5 | 80.6 | 81.2 |
| w/o action | 90.0 | 78.7 | 83.2 | 84.3 | 82.2 | 84.4 | 83.4 | 76.3 | 79.3 | 72.1 | 78.2 | 79.4 | 73 | 80.9 | 79.8 | 80.4 |
| w/o warm-start | 88.9 | 78.5 | 82.9 | 82.4 | 81.4 | 84.0 | 82.9 | 75.9 | 79.8 | 71.2 | 77.4 | 77.8 | 73.0 | 79.3 | 79.9 | 79.7 |
| only warm-start | 88.9 | 78.5 | 83.2 | 82.8 | 81.8 | 84.4 | 82.7 | 76.0 | 79.6 | 71.6 | 77.4 | 78.4 | 72.1 | 79.6 | 78.9 | 79.7 |
| w/o PNet | 88.8 | 78.1 | 82.9 | 82.8 | 81.6 | 83.9 | 82.7 | 75.7 | 79.0 | 71.4 | 77.0 | 77.7 | 71.4 | 78.8 | 78.7 | 79.4 |

Table 9: XNLI accuracy scores for each language.

| Lang. | en | de | es | fr | ja | ko | zh | AVG |
|---|---|---|---|---|---|---|---|---|
| XLM-R$_{large}$(Hu et al., 2020) | 94.7 | 89.7 | 90.1 | 90.4 | 78.7 | 79.0 | 82.3 | 86.4 |
| InfoXLM† | 95.1 | 89.0 | 91.0 | 92.2 | **86.1** | **85.4** | 86.6 | 89.3 |
| ERNIE-M† | 95.9 | 91.0 | 92.1 | 91.4 | 80.0 | 82.1 | 82.7 | 87.9 |
| ERNIE-M(Ouyang et al., 2021) | **96.0** | 91.9 | 91.4 | 92.2 | 83.9 | 84.5 | 86.9 | 89.5 |
| XY-LENT(Patra et al., 2022) | 95.5 | **92.3** | **92.5** | **93.2** | 84.0 | 83.7 | 86.7 | 89.7 |
| VECO2.0(Zhang et al., 2023) | 95.8 | 91.0 | 91.6 | 92.0 | 82.8 | 81.6 | 84.5 | 88.5 |
| Struct-XLM | **96.0** | 91.7 | 91.7 | 92.4 | 84.7 | 85.0 | **89.0** | **90.1** |
| w/o action | **96.0** | 91.3 | 91.3 | 92.1 | 84.6 | 85.0 | 85.6 | 89.4 |
| w/o warm-start | 95.9 | 92.2 | 91.7 | 92.5 | 82.4 | 82.8 | 85.5 | 89.0 |
| only warm-start | 95.6 | 91.2 | 91.8 | 91.9 | 82.4 | 81.6 | 84.7 | 88.5 |
| w/o PNet | 95.4 | 91.2 | 91.6 | 92.1 | 81.7 | 82.2 | 83.8 | 88.3 |

Table 10: PAWS-X accuracy scores for each language.

| Lang. | en | af | ar | bg | de | el | es | et | eu | fa | fi | fr | he | hi | hu | id | it |
|---|---|---|---|---|---|---|---|---|---|---|---|---|---|---|---|---|---|
| XLM-R$_{large}$(Hu et al., 2020) | 96.1 | **89.8** | 67.5 | 88.1 | 88.5 | 86.3 | 88.3 | 86.5 | 72.5 | 70.6 | 85.8 | 87.2 | 68.3 | 76.4 | 82.6 | 72.4 | 89.4 |
| VECO2.0(Zhang et al., 2023) | **96.2** | 89.4 | **70.0** | 88.6 | **89.7** | 86.6 | 89.0 | 87.2 | 75.1 | 71.4 | 86.1 | 87.7 | 70.2 | 74.7 | **84.2** | 72.8 | 89.8 |
| InfoXLM† | 96.0 | 88.6 | 69.2 | 88.1 | 88.9 | 87.3 | 88.4 | 85.7 | 74.5 | **71.6** | 85.8 | 87.1 | **71.9** | 69.3 | 83.1 | 72.7 | 88.3 |
| ERNIE-M† | 96.1 | 88.5 | 67.6 | 88.8 | 87.8 | 85.5 | 88.7 | 86.0 | 75.4 | 72.2 | 85.7 | 85.1 | 66.1 | 77.0 | 83.0 | 72.3 | 89.8 |
| Struct-XLM | 96.1 | 89.7 | 68.7 | **89.4** | 89.6 | **88.6** | 89.4 | **87.4** | **76.6** | 71.6 | **86.4** | 89.0 | 69.9 | 77.3 | 84.0 | 72.9 | 90.7 |
| w/o action | 96.1 | 88.9 | 69.3 | 89.0 | 89.2 | 86.9 | **89.9** | **87.4** | 76.1 | 71.2 | 86.2 | **89.4** | 65.5 | 73.4 | 83.0 | **73.3** | **91.0** |
| w/o warm-start | 96.0 | 89.7 | 69.6 | 88.9 | 88.9 | 87.6 | 89.2 | 86.9 | 72.7 | 71.0 | 86.1 | 88.3 | 70.4 | **77.7** | 83.4 | 72.9 | 89.7 |
| only warm-start | 95.8 | 88.5 | 67.5 | 88.3 | 89.1 | 87.1 | 88.6 | 86.7 | 74.2 | 70.5 | 85.3 | 87.9 | 66.8 | 72.9 | 82.3 | 72.9 | 89.9 |
| w/o PNet | 95.7 | 88.5 | 67.3 | 87.3 | 88.7 | 85.7 | 88.4 | 85.3 | 74.5 | 69.6 | 84.9 | 87.3 | 66.9 | 73.3 | 81.2 | 72.9 | 89.5 |

| | ja | kk | ko | mr | nl | pt | ru | ta | te | th | tl | tr | ur | vi | yo | zh | AVG |
|---|---|---|---|---|---|---|---|---|---|---|---|---|---|---|---|---|---|
| XLM-R$_{large}$(Hu et al., 2020) | 15.9 | 78.1 | 53.9 | 80.8 | 89.5 | 87.6 | 89.5 | 65.2 | 86.6 | 47.2 | 92.2 | 76.3 | 70.3 | 56.8 | 24.6 | 25.7 | 73.8 |
| VECO2.0(Zhang et al., 2023) | 36.2 | 78.3 | 53.4 | 84.7 | 89.8 | 88.8 | 89.8 | 64.9 | 84.5 | 50.9 | 93.3 | 76.8 | 67 | 58.8 | 23.2 | 40.7 | 75.4 |
| InfoXLM† | 38.9 | 78.0 | **54.3** | 82.9 | 89.3 | 88.0 | 89.2 | 62.2 | 86.6 | 56.7 | 92.9 | 76.2 | 58.7 | 59.2 | 26.7 | 59.2 | 75.6 |
| ERNIE-M† | **54.7** | 79.0 | 53.8 | 84.0 | 89.4 | 88.6 | 89.7 | 61.9 | 86.4 | **58.1** | 92.1 | **77.5** | 72.6 | 59.2 | 24.2 | **63.7** | 76.7 |
| Struct-XLM | 39.2 | **80.7** | 52.7 | **85.5** | **90.1** | **90.2** | **90.6** | **67.8** | 86.7 | 57.1 | **95.2** | 77.2 | **73.2** | 60.2 | **45.9** | 52.3 | **77.6** |
| w/o action | 34.1 | 79.7 | 51.8 | 84.2 | 89.3 | 88.5 | 90.2 | 66.2 | 87.0 | 53.2 | 93.1 | 76.5 | 70.4 | 59.5 | 40.7 | 39.7 | 76.1 |
| w/o warm-start | 38.3 | 79.2 | 53.4 | 83.6 | 89.7 | 88.7 | 90.1 | 64.1 | **86.9** | 54.4 | 94.0 | 76.7 | 73.0 | **60.3** | 26.1 | 51.4 | 76.3 |
| only warm-start | 30.2 | 78.7 | 50.4 | 84.5 | 89.2 | 88.1 | 89.7 | 65.5 | 85.2 | 51.5 | 94.1 | 76.0 | 67.1 | 58.8 | 33.1 | 37.1 | 75.0 |
| w/o PNet | 36.6 | 77.8 | 50.5 | 83.1 | 88.9 | 87.9 | 88.8 | 62.8 | 82.1 | 50.4 | 95.1 | 75.9 | 68.5 | 58.4 | 34.9 | 43.2 | 74.9 |

Table 11: POS results (Accuracy) for each language

| Lang. | en | af | ar | bg | bn | de | el | es | et | eu | fa | fi | fr | he | hi | hu | id | it | ja | jv |
|---|---|---|---|---|---|---|---|---|---|---|---|---|---|---|---|---|---|---|---|---|
| XLM-R_large(Hu et al., 2020) | 84.7 | 78.9 | 53.0 | 81.4 | 78.8 | 78.8 | 79.5 | 79.6 | 79.1 | 60.9 | 61.9 | 79.2 | 80.5 | 56.8 | 73.0 | 79.8 | 53 | 81.3 | 23.2 | 62.5 |
| VECO2.0(Zhang et al., 2023) | **84.8** | 78.5 | 51.8 | 81.3 | 79.2 | 80.7 | 80.7 | 75.8 | **82.4** | **69.6** | 66.3 | 81.1 | 80.7 | 56.2 | **73.1** | 82.9 | 54.1 | **82.1** | 18.8 | 66.1 |
| InfoXLM† | 84.6 | **81.0** | 57.3 | 83.6 | 77.1 | **81.1** | **81.5** | 81.6 | 78.9 | 68.5 | 53.5 | 81.5 | **82.2** | 56.9 | 72.9 | 81.7 | 54.4 | 82.7 | 28.7 | 66.1 |
| ERNIE-M† | 83.9 | 80.0 | 55.4 | 80.6 | 76.6 | 79.4 | 77.0 | 77.2 | 77.9 | 61.6 | 58.1 | 79.5 | 81.9 | 54.4 | 72.6 | 79.7 | **56.3** | 82.1 | 20.7 | 60.1 |
| Struct-XLM | 84.0 | 80.7 | **59.1** | **83.9** | **79.6** | 80.8 | 81.2 | 81.4 | 79.2 | 67.4 | **69.8** | **81.8** | 81.1 | **58.6** | 72.6 | 81.3 | 55.5 | 82.1 | **29.6** | **67.2** |
| w/o action | 84.0 | 79.2 | 54.4 | 82.5 | 77.0 | 78.8 | 79.4 | 79.4 | 78.6 | 64.3 | 66.4 | 80.7 | 81.0 | 58.2 | 71.0 | 80.3 | 55.3 | 81.4 | 27.0 | 63.8 |
| w/o warm-start | 82.8 | 78.1 | **59.1** | 79.3 | 73.8 | 77.9 | 77.8 | 73.5 | 79.6 | 61.9 | 62.6 | 79.9 | 78.8 | 54.1 | 70.8 | 79.2 | 52.6 | 81.7 | 20.1 | 62.8 |
| only warm-start | 82.6 | 80.0 | 47.2 | 80.9 | 75.4 | 78.6 | 77.3 | 77.6 | 79.2 | 67.3 | 64.1 | 80.7 | 79.6 | 50.9 | 69.4 | 80.2 | 54.9 | 81.5 | 19.2 | 65.3 |
| w/o PNet | 81.6 | 78.6 | 50.1 | 79.6 | 75.3 | 77.7 | 77.0 | 75.9 | 76.4 | 59.6 | 54.3 | 78.7 | 79.4 | 50.3 | 66.9 | 78.7 | 54.1 | 80.8 | 15.3 | 61.4 |

| Lang. | ka | kk | ko | ml | mr | ms | my | nl | pt | ru | sw | ta | te | th | tl | tr | ur | vi | yo | zh |
|---|---|---|---|---|---|---|---|---|---|---|---|---|---|---|---|---|---|---|---|---|
| XLM-R_large(Hu et al., 2020) | 71.6 | 56.2 | 60 | 67.8 | 68.1 | 57.1 | 54.3 | 84 | 81.9 | 69.1 | 70.5 | 59.5 | 55.5 | 1.3 | 73.2 | 76.1 | 56.4 | **79.4** | 33.6 | 33.1 |
| VECO2.0(Zhang et al., 2023) | **72.8** | 53.7 | 59.3 | 67.9 | 66.0 | 68.8 | 56.0 | 85.0 | 82.2 | 72.5 | 69.3 | **61.4** | 54.3 | 1.8 | 75.8 | **82.5** | **76.0** | 78.0 | 49.3 | 28.0 |
| InfoXLM† | 70.5 | 57.3 | 60.1 | 64.6 | 67.5 | 71.6 | **58.6** | **85.2** | **83.1** | **74.6** | **71.7** | 59.7 | 55.4 | 2.1 | 75.2 | 81.9 | 71.9 | 77.4 | 50.0 | **38.3** |
| ERNIE-M† | 69.5 | 55.4 | 56.7 | 66.1 | 67.9 | 70.1 | 57.2 | 83.9 | 81.6 | 71.2 | 70.7 | 61.3 | 54.3 | 2.9 | **76.0** | 80.1 | 66.9 | 76.8 | 45.1 | 30.4 |
| Struct-XLM | 72.0 | **56.7** | 59.7 | 68.2 | 68.9 | 74.3 | 58.1 | 84.7 | 83.0 | 72.6 | 70.5 | 60.6 | **56.8** | **3.0** | 75.9 | 81.8 | 70.7 | 78.2 | **58.3** | 32.9 |
| w/o action | 70.1 | 54.4 | **60.6** | 65.1 | 65.4 | 69.4 | 57.6 | 83.8 | 82.6 | 71.1 | 69.9 | 60.8 | 56.0 | 2.7 | 73.2 | 80.9 | 69.1 | 78.7 | 39.0 | 34.4 |
| w/o warm-start | 69.4 | 51.2 | 57.8 | 68.2 | 70.2 | 66.0 | 58.3 | 82.0 | 79.5 | 65.7 | 70.0 | 58.3 | 56.4 | 1.4 | 75.0 | 79.0 | 66.4 | 75.8 | 52.5 | 34.4 |
| only warm-start | 61.1 | 52.7 | 53.6 | 62.9 | 64.4 | 69.4 | 50.2 | 83.7 | 82.0 | 68.4 | 69.4 | 55.6 | 51.6 | 2.1 | 72.1 | 80.7 | 62.2 | 75.8 | 42.4 | 26.7 |
| w/o PNet | 60.5 | 50.8 | 52.7 | 62.5 | 60.4 | 68.7 | 50.4 | 83.5 | 81.6 | 66.2 | 68.9 | 54.3 | 50.5 | 2.1 | 70.8 | 79.7 | 61.2 | 75.5 | 41.4 | 22.6 |

Table 12: NER results (F1 Score) for each language

| Lang. | en | ar | de | el | es | hi | ru | th | tr | vi | zh | AVG |
|---|---|---|---|---|---|---|---|---|---|---|---|---|
| XLM-R_large(Hu et al., 2020) | 86.5/75.7 | 68.6/49.0 | 80.4/63.4 | 79.8/61.7 | 82.0/63.9 | 76.7/59.7 | 80.1/64.3 | 74.2/62.8 | 75.9/59.3 | 79.1/59.0 | 59.3/50.0 | 76.6/60.8 |
| XY-LENT(Patra et al., 2022) | 87.2/76.0 | 72.9/56.0 | 80.0/**64.5** | 79.6/63.5 | 81.2/63.1 | 75.3/59.7 | 77.7/61.5 | 70.9/59.5 | 74.0/58.7 | 77.4/59.2 | 69.0/**61.0** | 76.8/62.1 |
| VECO2.0(Zhang et al., 2023) | 88.2/**78.4** | 74.2/56.9 | 81.0/63.7 | **81.5**/**63.6** | 83.1/64.6 | 76.8/60.3 | 80.1/64.4 | 76.0/64.4 | 76.6/60.8 | 80.5/60.8 | **70.1**/60.9 | 78.9/63.7 |
| InfoXLM† | **88.7**/78.3 | 77.1/59.5 | 81.1/64.0 | **81.5**/63.4 | **83.8**/**65.5** | 77.7/60.7 | 81.2/**65.0** | 76.9/66.1 | **77.7**/60.8 | 80.4/61.1 | 66.2/57.1 | 79.3/63.8 |
| ERNIE-M† | 87.4/76.3 | 73.6/55.9 | 80.5/63.4 | 80.5/62.4 | 82.3/63.2 | **77.8**/**61.4** | 80.2/64.6 | 76.7/60.4 | 76.7/61.6 | 80.7/61.6 | 66.4/57.2 | 78.2/62.7 |
| Struct-XLM | 88.2/78.2 | **77.4**/**59.8** | **81.2**/61.9 | 81.0/63.2 | 83.3/65.1 | 77.7/60.2 | **81.4**/64.7 | **77.2**/**67.0** | 76.8/**61.2** | **81.0**/**62.2** | 69.2/59.2 | **79.5**/**64.1** |
| w/o action | 87.6/76.7 | 76.3/59.0 | 80.8/63.8 | 80.5/62.9 | 83.1/64.2 | 76.7/60.5 | 80.6/**65.0** | 76.3/65.5 | 76.2/60.2 | **81.0**/61.9 | 69.0/59.2 | 78.9/63.5 |
| w/o warm-start | 87.9/76.6 | 75.9/58.4 | 80.4/63.2 | 80.1/61.6 | 81.9/62.4 | 76.8/60.3 | 80.2/63.3 | 76.5/64.2 | 76.3/60.3 | 80.5/61.3 | 68.6/59.7 | 78.6/62.8 |
| only warm-start | 87.1/75.8 | 75.2/58.2 | 80.9/63.4 | 80.3/61.0 | 82.0/62.8 | 76.3/59.8 | 80.6/64.1 | 74.7/62.9 | 76.0/59.7 | 79.9/61.1 | 66.4/57.2 | 78.1/62.4 |
| w/o PNet | 87.1/76.3 | 75.4/58.0 | 80.0/62.1 | 80.0/62.8 | 81.2/62.4 | 75.0/58.2 | 79.4/62.8 | 73.4/63.0 | 75.0/59.0 | 80.4/61.3 | 66.1/54.3 | 77.5/61.8 |

Table 13: XQuAD results (F1 / EM) for each language.

| Lang. | en | ar | de | es | hi | vi | zh | AVG |
|---|---|---|---|---|---|---|---|---|
| XLM-R_large(Hu et al., 2020) | 83.5/70.6 | 66.6/47.1 | 70.1/54.9 | 74.1/56.6 | 70.6/53.1 | 74.0/52.9 | 62.1/37.0 | 71.6/53.2 |
| InfoXLM† | **84.7**/**71.7** | 67.5/47.3 | **71.7**/**56.9** | 74.9/56.9 | 72.2/54.0 | 74.8/53.6 | 68.7/45.3 | 73.5/55.1 |
| InfoXLM(Chi et al., 2021) | 84.5/71.6 | 67.6/47.6 | 71.2/56.2 | **75.1**/57.3 | 72.5/54.2 | **75.2/54.1** | 69.2/45.4 | 73.6/55.2 |
| ERNIE-M† | 83.7/70.8 | 65.5/46.3 | 70.5/55.7 | 74.0/56.1 | 71.3/53.6 | 74.2/52.7 | 70.2/46.7 | 72.8/54.6 |
| ERNIE-M(Ouyang et al., 2021) | 84.4/71.5 | 67.4/47.2 | 70.8/55.9 | 74.8/56.6 | **72.6/54.7** | 75.0/53.7 | **71.1/47.5** | **73.7/55.3** |
| XY-LENT(Patra et al., 2022) | 83.1/70.3 | 63.9/43.9 | 68.9/54.0 | 73.3/55.1 | 69.0/51.7 | 72.7/52.0 | 68.0/45.2 | 71.3/53.2 |
| VECO2.0(Zhang et al., 2023) | 84.1/71.4 | **74.3/56.3** | 70.3/54.9 | 66.5/46.5 | 71.5/53.7 | 74.2/53.1 | 67.9/43.7 | 72.7/54.2 |
| Struct-XLM | 84.4/71.6 | 66.8/46.4 | 71.4/55.9 | 74.9/**57.6** | 71.9/53.7 | 74.7/53.6 | 68.1/44.7 | 73.2/54.8 |
| w/o action | 84.1/70.8 | 66.7/46.3 | 70.7/55.3 | 74.4/56.9 | 71.5/53.4 | 74.3/52.6 | 66.4/44.3 | 72.6/54.2 |
| w/o warm-start | 83.9/70.7 | 66.1/45.5 | 71.0/55.3 | 74.3/56.2 | 71.9/53.7 | 73.9/52.8 | 62.9/43.6 | 72.0/54.0 |
| only warm-start | 83.7/70.4 | 66.0/45.1 | 70.4/55.2 | 74.2/56.0 | 71.2/52.6 | 74.1/52.6 | 62.5/44.4 | 71.7/53.8 |
| w/o PNet | 84.0/70.8 | 66.0/45.6 | 70.4/56.3 | 74.1/56.2 | 70.2/52.6 | 73.8/52.6 | 60.7/38.9 | 71.3/53.3 |

Table 14: MLQA results (F1 / EM) for each language.

| Lang. | en | ar | bn | fi | id | ko | ru | sw | te | AVG |
|---|---|---|---|---|---|---|---|---|---|---|
| XLM-R_large(Hu et al., 2020) | 71.5/56.8 | 67.6/40.4 | 64.0/47.8 | 70.5/53.2 | 77.4/61.9 | 31.9/10.9 | 67.0/42.1 | 66.1/48.1 | 70.1/43.6 | 65.1/45.0 |
| XY-LENT(Patra et al., 2022) | 73.4/59.1 | 71.6/54.1 | 63.7/51.3 | 66.5/52.3 | 77.0/63.4 | 57.2/43.5 | 68.0/**49.0** | 67.3/51.1 | 59.4/39.3 | 67.1/51.5 |
| VECO2.0(Zhang et al., 2023) | 73.0/60.7 | 73.7/55.8 | 63.6/46.9 | 72.8/56.9 | 79.6/66.7 | 60.5/47.8 | 67.7/39.9 | **73.2/60.1** | 75.6/**57.1** | 71.1/54.7 |
| InfoXLM† | **76.4**/63.9 | **75.6/57.0** | 67.1/52.2 | 71.3/55.1 | 79.9/63.4 | 61.0/49.3 | **69.2**/43.2 | 72.5/54.5 | **77.2**/52.5 | 72.2/54.6 |
| ERNIE-M† | 73.8/58.9 | 73.4/53.4 | 65.7/50.4 | 72.2/55.0 | 77.4/60.7 | 60.2/47.1 | 66.7/41.7 | 69.4/53.3 | 73.5/53.7 | 70.3/52.7 |
| Struct-XLM | 75.7/**64.5** | 75.3/56.0 | 69.2/**53.1** | 75.0/58.2 | 81.7/67.1 | **62.8/50.0** | 69.1/43.5 | 72.6/55.3 | 76.7/56.1 | **73.1/56.0** |
| w/o action | 73.5/60.9 | 74.9/53.7 | 66.3/47.8 | 73.7/57.0 | 80.8/66.5 | 61.9/**50.0** | 67.1/40.1 | 72.8/57.7 | 73.8/53.5 | 71.6/54.1 |
| w/o warm-start | 74.4/62.5 | 73.0/51.5 | **70.1**/52.2 | 73.8/56.3 | 80.2/63.5 | 61.4/47.8 | 66.6/40.1 | 72.8/54.5 | 74.3/55.0 | 71.8/53.7 |
| only warm-start | 72.5/59.1 | 73.9/54.2 | 65.3/50.4 | 72.7/56.1 | 80.0/64.2 | 60.5/47.5 | 65.9/38.8 | 72.6/55.5 | 73.7/49.5 | 70.8/52.8 |
| w/o PNet | 72.9/58.9 | 73.6/55.4 | 68.9/51.3 | 73.5/57.0 | 79.9/64.6 | 61.0/48.6 | 65.6/37.3 | 72.4/55.3 | 70.5/46.5 | 70.9/52.8 |

Table 15: TyDiQA-GoldP results (F1 / EM) for each language.