# OpenReview forum: "Struct-XLM: A Structure Discovery Multilingual Language Model for Enhancing Cross-lingual Transfer through Reinforcement Learning"
_EMNLP/2023/Conference — EMNLP 2023 Main_

### Official Review · Reviewer_xL1S · 2023-08-02

**Soundness:** 4

**Excitement:**

4: Strong: This paper deepens the understanding of some phenomenon or lowers the barriers to an existing research direction.

**Paper Topic And Main Contributions:**

This paper introduces a re-training RL procedure for multilingual LLMs, aimed at fostering their understanding of syntactic structures.
This procedure hinges on determining constituency-like structures to solve a cross-lingual translation ranking task.
Overall, the paper proposes a data-efficient way of re-training data-hungry LLMs..

**Questions For The Authors:**

1. Why apply the action matrix masking after the softmax in eq.3? doesn't that entail that weights E no longer sum to 1? Intuitively, I think this entails that you potentially have unbalanced attention outputs with very different norms?
2. Line 304, is "previous state" a typo, and did you mean all "previous states"? or are you only attending a single step?
3.  Lines 414-415: why not compare with a model with fewer parameters? you could verify that you observe the expected increase in performances in such a case.

**Reasons To Accept:**

- The proposed retraining approach is data-efficient,  requiring only a 8MB corpus (the authors contrast this with the 40GB+ requirements of concurrent methods).
- As far as I can tell, there seems to be significant novelty in the proposed approach, which successfully combines RL approaches, a translation ranking task and a Siamese network to  improve the cross-lingual capabilities of LLMs
- The authors conduct substantive experiments to verify their claims.

**Reasons To Reject:**

- In my opinion, the main advantage of the proposed approach over presented competitors is its low data requirements. This is however somewhat irrelevant, given that the approach is based on LLMs which are already data intensive. To a certain extent, a focus on verifying the viability of the proposed approach in low-resource scenarios would have been more coherent.
- Some claims are contradicted or not substantiated by the results provided by the authors: while the abstract and introduction suggest that the model "learns universal syntactic structures", the qualitative analysis reported in section 5.4 does not seem to match with expert syntactic annotations. If the authors intend to make this claim, then supplementary experiments would be required (e.g., measure the learned constituency structure against existing constituency benchmarks).
- Despite the numerous experiments conducted by the authors, the low margin of improvements would have call for statistical significance testing: as the paper currently stands, I believe there is a possibility that the reported results correspond to overfitting to the standard test splits of the cross-lingual benchmarks proposed here [(Gorman & Bedrick, 2019)](https://aclanthology.org/P19-1267/).
- The limitations section reads more as a series of possible future works rather than a thorough self-critique.

**Reproducibility:**

4: Could mostly reproduce the results, but there may be some variation because of sample variance or minor variations in their interpretation of the protocol or method.

**Reviewer Confidence:**

3: Pretty sure, but there's a chance I missed something. Although I have a good feel for this area in general, I did not carefully check the paper's details, e.g., the math, experimental design, or novelty.

**Typos Grammar Style And Presentation Improvements:**

- you use T as a scalar exponent in a few instances (lines 212 qand 213, in particular); Please use lowercase nonbold for scalar indices, and especially avoid T exponents as they can easily be mixed up with transpose operations.
- consider providing explicit formulas for computing the action matrix from the action vector. You are essentially computing $\mathbf{1} - \mathrm{Id}$ for every pseudo-constituent (with $\mathbf{1}$ a square matrix of ones); combining these in a block-matrix notation might allow you to provide a more formal overview of the computations you propose.
- please check the manuscript for typos and ungrammatical / confusing sentences. Some examples:
  + line 421: "Natrual" instead of "Natural"
  + lines 149--151: " [...] proposed a cross-attention module that builds interdependence between languages in model inner" ; I'm uncertain what you mean by "model inner"
  + line 356 "Refer to Zhang et al (2018), our training process consists of three steps:"  -> you probably mean "Similar to Zhang et al. ..."
  + line 549: "Qualitatively Analyze": prefer nominal titles, e.g. "Qualitative Analysis"

---

> ### Author Rebuttal · Authors · 2023-08-29
>
> We would like to express our sincere appreciation for your time and effort in reviewing our paper. Your expertise and thoughtful feedback have been invaluable in improving the quality of our work.
>
> - **R1: About the performance of the proposed approach in low-resource scenarios.**
>
> Thank you very much for acknowledging the data efficiency of our approach and for your valuable feedback. We will include an analysis of the improvements in low-resource language performance in the new version, which was omitted in the previous version due to space constraints. For instance, in Table 10, eu, tl, and yo do not have training corpora, yet they exhibit significant improvements in POS tasks compared to the best baseline result, with increases of 1.2%, 1.9%, and 16.0%, respectively. Moreover, in Table 11, fa and ms without warm-start training corpora improve by 3.5% and 2.7% than the best baseline result.
>
> - **R2: About quantitative analysis of syntax discovery.**
>
> I'm sorry that we accidentally used the absolute expression "learns universal syntactic structures". The expression "learning universal structural information" in line 79 is more in line with our original intention. More specifically, the structure information we learn is a coarse-grained constituent boundary information, namely action vector. It predicts constituent boundaries, thereby segmenting sentences into constituents. We will revise our representation in the new version and add evaluation results of constituent boundary prediction to quantify our learned structure. The labeled test data comes from the warm-start step.
>
> The results are shown in Table-a below, the boundary predicted by Struct-XLM can be 66.84% and 57.58% recall in English and target languages, respectively, which means Struct-XLM can correctly predict $>55\%$ constituents’ boundary in labeled test sets in target languages. Struct-XLM tends to predict more constituents (an average of 11-12 constituents in the target language sentence and the average in the test set is 6-7), as we mentioned in Lines 558-561. This may be the reason for the low precision (P). We also report the results of the model w/o warm-start to prove the influence of the warm-start step for structural discovery.
>
> Table a. The precision(P), recall(R), and F1 scores of the constituent boundary predict
>
> | **Models**     | **P**$_{en}$ | **R**$_{en}$ | F1$_{en}$ | **P**$_{tgt}$ | **R**$_{tgt}$ | F1$_{tgt}$ |
> | -------------- | -------------- | -------------- | ----------- | --------------- | --------------- | ------------ |
> | Struct-XLM     | **34.49**      | **66.84**      | **45.50**   | **35.67**       | **57.58**       | **43.14**    |
> | w/o warm-start | 30.57          | 62.13          | 40.98       | 31.63           | 56.48           | 39.67        |
>
> - **R3: About the statistical significance score of Struct-XLM in comparison to info-XLM.**
>
> We conducted significance score calculations on the dataset where our model achieved the best performance in Table 1, as well as the average of seven tasks in the XTREME benchmark (Task$_{AVG}$), as shown in the Table below. When compared to the InfoXLM model, we did not observe significant differences. However, our model achieved similar results using only 1/5000 training data of InfoXLM. My explanation in lines 444-449 is inaccurate in this regard, we should be competitive with but not superior to the InfoXLM model. We will make the necessary adjustments to the relevant experimental analysis in the new version.
>
> Table b. The statistical significance score.
>
> | **Models**                  | **PAWS-X** | **POS** | **NER** | **XQuAD(F1)** | **XQuAD(EM)** | **TyDiQA(F1)** | **TyDiQA(EM)** | **Task**$_{AVG}$ |
> | --------------------------- | ---------- | ------- | ------- | ------------- | ------------- | -------------- | -------------- | ------------------ |
> | XLM-Rlarge(Hu et al., 2020) | <0.01      | <0.01   | <0.01   | <0.05         | <0.05         | <0.05          | <0.05          | <0.01              |
> | XY-LENT(Patra et al., 2022) | 0.40       | \--     | \--     | <0.01         | <0.05         | <0.01          | <0.05          | <0.05              |
> | VECO2.0(Zhang et al., 2023) | <0.05      | <0.05   | <0.05   | 0.09          | 0.32          | <0.05          | 0.26           | <0.01              |
> | ERNIE-M$^\dagger$         | <0.05      | 0.47    | <0.01   | <0.05         | <0.05         | <0.01          | <0.01          | <0.01              |
> | InfoXLM$^\dagger$         | 0.19       | 0.08    | 0.37    | 0.36          | 0.35          | 0.15           | 0.10           | 0.22               |
>
> Since this benchmark is publicly available, and each dataset has official, standardized splits, further experiments are required to investigate whether overfitting may occur due to data partitioning. We believe this is an interesting direction, as the choice of data in a multilingual setting can indeed impact cross-lingual performance.
>
> - **R4: About limitations.**
>
> We attempted to analyze our limitations in terms of data, method, resource constraints, and time constraints. Moreover, we proposed some actions to improve. We appreciate the guidance from the reviewer, and We will carefully consider distinguishing the sections on limitations and future work more clearly, as per your suggestion.
>
> - **Q1: About the design of action matrix masking after the SoftMax in eq.3.**
>
> In the source code of the Transformer, there is a dropout operation applied to the attention matrix, which causes the attention matrix not to sum to 1. Given this, we directly use the action matrix after the SoftMax operation, like a dropout operation. We have also observed that after adding the action matrix, applying SoftMax normalization again may not be suitable because the self-attention weights are relatively small, and the parts with action matrix values of 0 would be assigned relatively large weights. We attempted to modify the 0 values in the action matrix to a very small number like 1e-9, but this led to underflow issues when multiplied by 1e-9 due to the very small weights in the padding position. Consequently, we opted for the computation method illustrated in Figure 2.
>
> - **Q2: About "previous state" in Line 304.**
>
> As described in Lines 83-85, we formulate the structure discovery as a sequential decision problem, so we only attend to a single step. It's possible that our previous explanation lacked detail and caused confusion. In addition to this clarification, we will revise the manuscript based on the modification suggestions you provided later on. We greatly appreciate your corrections.
>
> - **Q3: About the question of why it is not compared with the Syntax-MBERT model.**
>
> Comparing our model to the Syntax-MBERT model may not be entirely fair due to the differences in the number of model parameters. In fact, the results reported in the Syntax-MBERT paper were even lower than our XLM-R baseline. Considering the distinctions between our approach and Syntax-MBERT, we have not replicated the Syntax-MBERT model based on XLM-R or implement our model based on mBERT for comparison. Two key differences are that they explicitly use syntactic labels, and they utilize syntactic annotation not only for pre-training syntactic representations but also for fine-tuning each task.
>
> Of course, as you mentioned, if surpassing the Syntax-MBERT model based on XLM-R would provide stronger evidence, we have initiated efforts to replicate it.
>
> - **About the suggestion of presentation improvements.**
>
> For the conversion of the action vector to an action matrix, a more formal description is indeed needed. we provide a rough algorithm for computing the action matrix from the action vector. The final version will be added to the new manuscript.
>
> ---
>
> Algorithm: Convert the action vector to the action matrix
>
> ---
>
> Input: action vector $a$, length of sentence $L$
>
> Output: action matrix $A$
>
> 1. Initialize action matrix $A$ as a zero matrix with dimension $L$.
> 2. Iterate over the vector $a$ to get the start and end index of each constituent, save in a list $C=[[c_1^{start}, c_1^{end}]...[c_i^{start}, c_i^{end}]...[c_n^{start}, c_n^{end}]]$, where $c_i$ represents a constituent.
> 3. Iterate over the list $C$:
>
> If $c_i^{start}=c_i^{end}$, $c_i$ consists of single word, and A[i][i]=1,
>
> Else sub-action matrix $A^{c_i}$ is a matrix with a value of 1 except that the diagonal is 0, and the dimension is the length of constituents $c_i$. The starting position of $A^{c_i}$ in $A$ is $(c_i^{start}, c_i^{start})$
>
> 4. Output the action matrix $A$
>
> ---
>
> Thank you again for your helpful comments and suggestions on our work. Looking forward to further communication.

---

### Official Review · Reviewer_Jy7e · 2023-08-05

**Typos Grammar Style And Presentation Improvements:** Some spaces missing between text and …
**Soundness:** 4

**Excitement:**

4: Strong: This paper deepens the understanding of some phenomenon or lowers the barriers to an existing research direction.

**Paper Topic And Main Contributions:**

The paper introduces a novel multilingual language model that discovers structure and aligns representations across languages better than competing models. The approach incorporates reinforcement learning with binary actions to discover constituent structure. In the experimental section, the authors show that the model is very competitive and improves cross-lingual transfer even though it is trained on rather small datasets.

**Questions For The Authors:**

Is it possible to run the training process without warm-start or how much does a reasonable training process depend on a reasonable initialization? Would it be possible to quantify the changes that RL discovered compared to given structures in annotated data?
Is it necessary to have warm-start data in all languages that are included?
How important is batch-size optimization for the in-batch labeling for the contrastive training objective?

**Reasons To Accept:**

The paper is very well written and the approach is novel and effective. The idea of using reinforcement learning to discover structure based on translation ranking loss as a reward is interesting and seems to work well to align source and target language representations.

**Reasons To Reject:**

I don't see immediate reasons to reject the paper.

**Reproducibility:**

3: Could reproduce the results with some difficulty. The settings of parameters are underspecified or subjectively determined; the training/evaluation data are not widely available.

**Reviewer Confidence:**

3: Pretty sure, but there's a chance I missed something. Although I have a good feel for this area in general, I did not carefully check the paper's details, e.g., the math, experimental design, or novelty.

---

> ### Author Rebuttal · Authors · 2023-08-29
>
> We would like to begin by expressing our sincere gratitude to you for your time and expertise in evaluating our paper. In your review, you raised several important points that we would like to address.
>
> - **Q1: About the ablation without warm-start step**
>
> We deeply apologize for the absence of an ablation experiment without the warm-start step in our initial submission. We have conducted this experiment and reported the results in the table below. According to the results, the model without the warm-start step performs close to the model w/o action vector to inference and outperforms other ablation models and the PLM baseline, demonstrating the effectiveness of our approach. However, it doesn't surpass our final Struct-XLM model, confirming the effectiveness of our warm-start procedure. We will include these new results and analyses in the next version.
>
> Table a. Ablation results on each model.
>
> | **Models**      | **XNLI** | **PAWS-X** | **POS**  | **NER**  | **XQuAD(F1)** | **XQuAD(EM)** | **MLQA(F1)** | **MLQA(EM)** | **TyDiQA(F1)** | **TyDiQA(EM)** | **AVG**  |
> | --------------- | -------- | ---------- | -------- | -------- | ------------- | ------------- | ------------ | ------------ | -------------- | -------------- | -------- |
> | XLM-R           | 79.2     | 86.4       | 73.8     | 65.4     | 76.6          | 60.8          | 71.6         | 53.2         | 65.1           | 45.0           | 67.7     |
> | Struct-XLM      | **81.2** | **89.8**   | **76.8** | **68.0** | **79.4**      | **63.9**      | **73.0**     | **54.6**     | **72.9**       | **55.4**       | **71.5** |
> | w/o action      | 80.4     | 89.4       | 76.1     | 66.7     | 78.9          | 63.5          | 72.6         | 54.2         | 71.6           | 54.1           | 70.7     |
> | w/o warm-start  | 79.7     | 89.0       | 76.3     | 65.6     | 78.6          | 62.8          | 72.0         | 54.0         | 71.8           | 53.7           | 70.4     |
> | only warm-start | 79.7     | 88.5       | 75.0     | 64.5     | 78.1          | 62.4          | 71.7         | 53.8         | 70.8           | 52.8           | 69.7     |
> | w/o PNet        | 79.4     | 88.3       | 74.9     | 62.9     | 77.5          | 61.8          | 71.3         | 53.3         | 70.9           | 52.8           | 69.3     |
>
> - **Q2: About quantitative analysis of the changes that RL discovered compared to given structures in annotated data**
>
> As our structural discovery is unsupervised, we indirectly confirm that RL can discover structures better suited for cross-lingual alignment via the sentence retrieval task in Section 5.3. In response to your suggestion, we quantitatively assessed the constituent boundary (i.e. number 1 in the action vector) predicted by PNet. The labeled test set is from the warm-start step, including English (en) and target languages (tgt) two-way. The results are shown in Table-b below, the boundary predicted by Struct-XLM can be 66.84% and 57.58% recall in English and target languages, respectively, which means Struct-XLM can correctly predict $>55\%$ constituents’ boundary in labeled test sets in target languages. Struct-XLM tends to predict more constituents (an average of 11-12 constituents in the target language sentence and the average in the test set is 6-7), as we mentioned in Lines 558-561. This may be the reason for the low precision (P). We also report the results of the model w/o warm-start to prove the influence of the warm-start step for structural discovery.
>
> Table b. The precision(P), recall(R), and F1 scores of the constituent boundary predict
>
> | **Models**     | **P**$_{en}$ | **R**$_{en}$ | F1$_{en}$ | **P**$_{tgt}$ | **R**$_{tgt}$ | F1$_{tgt}$ |
> | -------------- | -------------- | -------------- | ----------- | --------------- | --------------- | ------------ |
> | Struct-XLM     | **34.49**      | **66.84**      | **45.50**   | **35.67**       | **57.58**       | **43.14**    |
> | w/o warm-start | 30.57          | 62.13          | 40.98       | 31.63           | 56.48           | 39.67        |
>
> - **Q3: About the necessary of warm-start data in all languages**
>
> It's not necessary for all languages to have warm-start data. For instance, in Table 10, eu, tl, and yo do not have training corpora, yet they exhibit significant improvements in POS tasks compared to the best baseline result, with increases of 1.2%, 1.9%, and 16.0%, respectively. Moreover, in Table 11, fa and ms without warm-start training corpora improve by 3.5% and 2.7% than the best baseline result. We will incorporate analysis in the new version regarding the performance improvements in low-resource languages that lack warm-starting data.
>
> - **Q4: About the importance of batch-size setting**
>
> Our goal is to bring parallel sentences closer and non-parallel sentences farther apart. However, since our training dataset only consists of parallel sentence pairs, we adopted a batch sampling approach to construct comparable data containing non-parallel sentences. The setting of batch size is unlimited, and we think that a larger batch size should be able to better distinguish parallel data from non-parallel data. However, due to the particularity of RL training, we recommend setting the value smaller, and we use 5.
>
> Thank you again for acknowledging the writing and methodology of our work. We look forward to further discussions.

---

### Official Review · Reviewer_eQtC · 2023-08-09

**Soundness:** 4

**Excitement:**

4: Strong: This paper deepens the understanding of some phenomenon or lowers the barriers to an existing research direction.

**Paper Topic And Main Contributions:**

The paper introduces a novel multilingual pre-trained language model (mPLM) named Struct-XLM. This model aims to enhance cross-lingual alignment and subsequent cross-lingual transfer by incorporating structural information into the existing mPLM, specifically XLM. This structural information is automatically extracted through a reinforcement learning (RL) algorithm.

The architectural design of the model comprises three key components: the Policy Network (PNet), the Structural Encoder, and the Translation Ranking Task module. The Structural Encoder (mPLM) serves as an environment, furnishing state representations utilized by PNet to generate action vectors corresponding to sentence structures.  This action vector is used with a multi-head attention module in the last layer of mPLM, and Translation Ranking Task to provide a delayed reward to PNet. Empirical evaluations are conducted on the XTREME benchmark across seven tasks, and the paper claims improved performance across these tasks. The literature review section is thoroughly written and appreciated, and relevant works are cited. The Paper is compared with several baseline models including, Info-XLM, ERNIE-M, XY-LENT and VECO2.0.

**Questions For The Authors:**

(A) What is the statistical significance score of Struct-XLM in comparison to info-XLM?
(B) How good were the extracted structures?
(C) What is the rationale behind mapping the action vector to the action matrix? Have ablations been conducted when directly using the action vector does not yield improvements?
(D) Additional ablation is needed: What are the consequences if the warm-start step is omitted? Is there no benefit from the multilingual representation?

**Reasons To Accept:**

(1) The paper presents a novel contribution by automatically discovering syntactic structures. This holds the potential to bring benefits to numerous low-resource languages that encounter limitations in labeled datasets.
(2) Integrating the syntactic structure to enhance cross-lingual alignment and expand language coverage is intuitive. On a theoretical level, the paper is supported by relevant and well-incorporated literature references. I do not have any concerns in this regard.
(3) The paper is well written. The mode performance is compared with state-of-the-art baselines.

**Reasons To Reject:**

(1) The paper In its current form, the methodology section of the paper presents a series of steps conducted to perform an experiment and attain results. However, it falls short in conveying the underlying intuition or motivations driving each of these steps, which is crucial for establishing a connection with the reader. For instance, in line 214, the authors assert that "The action vector can represent the structure of the sentence," but this assertion lacks elaboration in sections 3.2 where the underlying motivation should be articulated. A similar observation applies to sections 3.3 and 3.4.
(2) I view this model as an alternative competitive approach with info-XLM, rather than superior. Observing Table-1, 2, and 4, it becomes apparent that the performance improvement is not notably substantial and might not possess statistical significance. The authors should provide the statistical significance scores to ensure a more robust conclusion. Given the results, the paper's claims could be adjusted from asserting superiority to positioning the model as competitive. The paper in it’s current form lacks this type of analysis.
(3) The paper lacks a detailed analysis of the discovered structures. While there are efforts made in section 5.4, they are not very convincing. Since automated structure extraction is the main claim of the paper (ultimately contributing to improved cross-lingual alignment and transfer), I believe it should be the primary focus. A potential direction could involve establishing quantitative similarities or correlations between the discovered structures and human-extracted structures using a small dataset.
(4) Another limitations of the work is the utilization of a 13.7k parallel corpus for warm-start training. This essentially restricts the model's applicability to a smaller set of low-resource languages. There should be an analysis or experimentation conducted to overcome this limitation.

**Reproducibility:**

3: Could reproduce the results with some difficulty. The settings of parameters are underspecified or subjectively determined; the training/evaluation data are not widely available.

**Reviewer Confidence:**

4: Quite sure. I tried to check the important points carefully. It's unlikely, though conceivable, that I missed something that should affect my ratings.

---

> ### Author Rebuttal · Authors · 2023-08-29
>
> Thank you very much for acknowledging our contributions to cross-lingual alignment and low-resource language research with the idea of introducing syntactic structure. We appreciate your feedback and the time you've taken to evaluate our work. Based on your comments, we would like to make a response and hope for your further comments.
>
> - **R1: About the motivation statement for each step of our method.**
>
> In this version, we emphasized explaining the methodology but didn't describe deeply the underlying motivations in section 3. We appreciate your feedback as a reader for pointing this out, and we will enhance it in the new version.
>
> In lines 245-249, we explained how the action vector can represent the structure of the sentence, which provides information about the constituent structure. In the new manuscript, we'll add a syntactic parse tree example for better understanding. This parse tree can be projected to the sequence level to identify constituents, as shown in Figure 2: 'This gives them $I$ a competitive edge $|$ for the interim period $|$.' It segments the sentence into three constituents, excluding punctuation. Thus, the action vector predicts sentence constituent boundaries. In Section 3.3, we'll explain our choice of the translation ranking task, known to enhance multilingual representation. As for PNet's motivation, we discussed it in lines 83-87 and will emphasize it in Section 3.4.
>
> - **QC: About the mapping from the action vector to the action matrix.**
>
> The action vector represents sentence constituent boundaries, but each word needs a sub-action vector to focus on other words in the same component. Hence, we'll convert the action vector to the action matrix. They essentially convey the same information, eliminating the need for ablation.
>
> We acknowledge that our unclear previous explanation in lines 245-249 may cause confusion. We will revise our description. Additionally, we can provide a rough algorithm for computing the action matrix from the action vector. The final version will be added to the new manuscript.
>
> ---
>
> Algorithm: Convert the action vector to the action matrix
>
> ---
>
> Input: action vector $a$, length of sentence $L$
>
> Output: action matrix $A$
>
> 1. Initialize action matrix $A$ as a zero matrix with dimension $L$.
> 2. Iterate over the vector $a$ to get the start and end index of each constituent, save in a list $C=[[c_1^{start}, c_1^{end}]...[c_i^{start}, c_i^{end}]...[c_n^{start}, c_n^{end}]]$, where $c_i$ represents a constituent.
> 3. Iterate over the list $C$:
>
> If $c_i^{start}=c_i^{end}$, $c_i$ consists of single word, and A[i][i]=1,
>
> Else sub-action matrix $A^{c_i}$ is a matrix with a value of 1 except that the diagonal is 0, and the dimension is the length of constituents $c_i$. The starting position of $A^{c_i}$ in $A$ is $(c_i^{start}, c_i^{start})$
>
> 4. Output the action matrix $A$
>
> ---
>
> - **R3QB: About quantitative analysis of syntax discovery.**
>
> As we aim to acquire structural information that facilitates cross-lingual alignment, we employed retrieval tasks in Section 5.3 to quantitatively analyze whether the information obtained through RL contributes to cross-lingual alignment.
>
> In response to your suggestion, we try to conduct a quantitative analysis of the discovered structure by evaluating the constituents' boundary predicted by the action vector. The labeled test set is from the warm-start step, including English (en) and target languages (tgt) two-way. The results are shown in Table-a below, the boundary predicted by Struct-XLM can be 66.84% and 57.58% recall in English and target languages, respectively, which means Struct-XLM can correctly predict $>55\%$ constituents’ boundary in labeled test sets in target languages. Struct-XLM tends to predict more constituents (an average of 11-12 constituents in the target language sentence and the average in the test set is 6-7), as we mentioned in Lines 558-561. This may be the reason for the low precision (P). We also report the results of the model w/o warm-start to prove the influence of the warm-start step for structural discovery.
>
> Table a. The precision(P), recall(R), and F1 scores of the constituent boundary prediction.
>
> | **Models**     | **P**$_{en}$ | **R**$_{en}$ | F1$_{en}$ | **P**$_{tgt}$ | **R**$_{tgt}$ | F1$_{tgt}$ |
> | -------------- | -------------- | -------------- | ----------- | --------------- | --------------- | ------------ |
> | Struct-XLM     | **34.49**      | **66.84**      | **45.50**   | **35.67**       | **57.58**       | **43.14**    |
> | w/o warm-start | 30.57          | 62.13          | 40.98       | 31.63           | 56.48           | 39.67        |
>
> - **R2QA: About the statistical significance score of Struct-XLM in comparison to info-XLM.**
>
> We conducted significance score calculations on the dataset where our model achieved the best performance in Table 1, as well as the average of seven tasks in the XTREME benchmark (Task$_{AVG}$), as shown in Table-b below. In the average of the XTREME benchmark, the statistical significance score of Struct-XLM in comparison to info-XLM is 0.22. **As you mentioned, we should be competitive but not superior with the InfoXLM model, achieving similar results using only 1/5000 training data of InfoXLM.** My analysis in lines 444-449 is inaccurate in this regard. We will make the necessary adjustments to the discourse of the relevant experimental analysis in the new version.
>
> Table b. The statistical significance score.
>
> | **Models**                  | **PAWS-X** | **POS** | **NER** | **XQuAD(F1)** | **XQuAD(EM)** | **TyDiQA(F1)** | **TyDiQA(EM)** | **Task**$_{AVG}$ |
> | --------------------------- | ---------- | ------- | ------- | ------------- | ------------- | -------------- | -------------- | ------------------ |
> | XLM-Rlarge(Hu et al., 2020) | <0.01      | <0.01   | <0.01   | <0.05         | <0.05         | <0.05          | <0.05          | <0.01              |
> | XY-LENT(Patra et al., 2022) | 0.40       | \--     | \--     | <0.01         | <0.05         | <0.01          | <0.05          | <0.05              |
> | VECO2.0(Zhang et al., 2023) | <0.05      | <0.05   | <0.05   | 0.09          | 0.32          | <0.05          | 0.26           | <0.01              |
> | ERNIE-M$^\dagger$         | <0.05      | 0.47    | <0.01   | <0.05         | <0.05         | <0.01          | <0.01          | <0.01              |
> | InfoXLM$^\dagger$         | 0.19       | 0.08    | 0.37    | 0.36          | 0.35          | 0.15           | 0.10           | 0.22               |
>
> - **QD: About additional ablation.**
>
> Thank you for bringing this to our attention. We unintentionally missed evaluating the ablation model without the warm-start step (w/o warm-start). We've now conducted this experiment and added the results in the table below. According to the results, the model w/o warm-start performs close to the model w/o an action vector to inference and outperforms other ablation models and the PLM baseline, demonstrating the effectiveness of our approach. However, it doesn't surpass our final Struct-XLM model, confirming the effectiveness of our warm-start procedure. We will include these new results and analyses in the next version.
>
> Table c. Ablation results on each model. The results except the model w/o warm-start is new, others are from Table 3.
>
> | **Models**      | **XNLI** | **PAWS-X** | **POS**  | **NER**  | **XQuAD(F1)** | **XQuAD(EM)** | **MLQA(F1)** | **MLQA(EM)** | **TyDiQA(F1)** | **TyDiQA(EM)** | **AVG**  |
> | --------------- | -------- | ---------- | -------- | -------- | ------------- | ------------- | ------------ | ------------ | -------------- | -------------- | -------- |
> | XLM-R           | 79.2     | 86.4       | 73.8     | 65.4     | 76.6          | 60.8          | 71.6         | 53.2         | 65.1           | 45.0           | 67.7     |
> | Struct-XLM      | **81.2** | **89.8**   | **76.8** | **68.0** | **79.4**      | **63.9**      | **73.0**     | **54.6**     | **72.9**       | **55.4**       | **71.5** |
> | w/o action      | 80.4     | 89.4       | 76.1     | 66.7     | 78.9          | 63.5          | 72.6         | 54.2         | 71.6           | 54.1           | 70.7     |
> | w/o warm-start  | 79.7     | 89.0       | 76.3     | 65.6     | 78.6          | 62.8          | 72.0         | 54.0         | 71.8           | 53.7           | 70.4     |
> | only warm-start | 79.7     | 88.5       | 75.0     | 64.5     | 78.1          | 62.4          | 71.7         | 53.8         | 70.8           | 52.8           | 69.7     |
> | w/o PNet        | 79.4     | 88.3       | 74.9     | 62.9     | 77.5          | 61.8          | 71.3         | 53.3         | 70.9           | 52.8           | 69.3     |
>
> - **R4: About the limitation from parallel data for warm-start training.**
>
> Based on the experimental results in Table 8-14, even languages without warm-start training data can show improvements. For instance, in Table 10, eu, tl, and yo do not have training corpora, yet they exhibit significant improvements compared to the best baseline result, with increases of 1.2%, 1.9%, and 16.0%, respectively. Moreover, in Table 11, fa and ms without warm-start training corpora improve by 3.5% and 2.7% than the best baseline result. However, for more low-resource languages (such as languages not included in the PLM pre-training set), we have no data to prove whether our method is effective. We will supplement the relevant experimental analysis and limitation analysis in the new version.
>
> Your comments have been instrumental in improving the quality of our paper, and we are thankful for your expertise.

---

### Meta-Review · Area_Chair_PHYx · 2023-09-20

**Recommendation:** 5

**Metareview:**

The paper introduces a new multilingual LLM, Struct-XLM - which understand the underlying structure and aligns the representations across languages. The approach uses RL mechanism to discover the structural information using a translation ranking task.

The paper proposes a novel mechanism to discover structural information and uses it bring alignment across langauges. This approach is quite useful for low resource languages.
The paper is motivated well with detailed explanation and experimentation to verify their claims

Authors should update the draft with the discussion and additional details shared during the rebuttal phase.

---

### Decision · Program_Chairs · 2023-10-07

**Decision:**

Accept-Main

**Comment:**

The paper introduces a new multilingual LLM, Struct-XLM - which understand the underlying structure and aligns the representations across languages. The approach uses RL mechanism to discover the structural information using a translation ranking task.

The paper proposes a novel mechanism to discover structural information and uses it bring alignment across langauges. This approach is quite useful for low resource languages.
The paper is motivated well with detailed explanation and experimentation to verify their claims

Authors should update the draft with the discussion and additional details shared during the rebuttal phase.